# Quantifying biomolecular organisation in membranes with brightness-transit statistics

Falk Schneider [1,2] ✉, Pablo F. Cespedes [1], Narain Karedla[1,3],
Michael L. Dustin [1] & Marco Fritzsche [1,3] ✉

Cells crucially rely on the interactions of biomolecules at their plasma membrane to maintain homeostasis. Yet, a methodology to systematically quantify biomolecular organisation, measuring diffusion dynamics and oligomerisation, represents an unmet need. Here, we introduce the brightness-transit statistics (BTS) method based on fluorescence fluctuation spectroscopy and combine information from brightness and transit times to elucidate biomolecular diffusion and oligomerisation in both cell-free in vitro and in vitro systems incorporating living cells. We validate our approach in silico with computer simulations and experimentally using oligomerisation of EGFP tethered to supported lipid bilayers. We apply our pipeline to study the oligomerisation of CD40 ectodomain in vitro and endogenous CD40 on primary B cells. While we find a potential for CD40 to oligomerize in a concentration or ligand depended manner, we do not observe mobile oligomers on B cells. The BTS method combines sensitive analysis, quantification, and intuitive visualisation of dynamic biomolecular organisation.

Quantifying the interactions of biomolecules is important for the understanding of cellular function, decision making and behaviour. Over the past decades, a large body of groundbreaking work from ultra-fast single molecule tracking, to advanced image correlation approaches, and super-resolved dynamic measurements have provided tremendous insights into plasma membrane organisation[1]. Yet, these methodologies often require expert knowledge, necessitate specialised hardware and analysis tools, and/or have limited sensitivity to molecular oligomerisation[1–4]. Nonetheless, being able to assess diffusion dynamics and molecular oligomerisation simultaneously is key to understanding the inner workings of living cells and their interactions with the environment. Changes in organisation of lipids, lipid anchored signalling proteins, and receptors in the plasma membrane governs many cellular processes and is known to critically influence the activation of immune cells and their ability to respond to pathogenic threats[5–8]. Assembly of T-cell receptor (TCR) monomers into TCR oligomers, generally referred to as nano- and micro-clusters depending on size or methodology, represent a hallmark of T-cell

activation and is crucial to initiate large-scale morphological changes such as the formation of the immunological synapse (IS)[5,9]. This process is further accompanied by reorganisation of various co-receptors at the membrane, constantly adapting to physiological needs. An important signal in the activation of T cells involves interaction of the T-cell associated CD40 ligand (CD40LG) and its counterpart, the Tumour Necrosis Family (TNF) Receptor superfamily member CD40 expressed on professional antigen presenting cells (APCs) such as B-cells[10]. CD40 is a key receptor for T cell dependent antibody responses and it has been therapeutically targeted by agonistic antibodies in multiple types of cancer[11,12]. For example, CD40 has been indicated to play a role in the immune response in bladder cancer when targeted by antibodies on dendritic cells[13]. In the membrane of activating T cells, the functional form of CD40LG is a homo-trimer potentially oligomerising the CD40 receptors on the APC[14,15]. The oligomerisation of CD40 receptors in APCs can also be induced in vitro with soluble anti-CD40 antibodies binding near the CD40LG binding site[16,17], highlighting the potential importance of CD40 oligomerisation

[1]Kennedy Institute for Rheumatology, Roosevelt Drive, University of Oxford, Oxford OX3 7LF, United Kingdom. [2]Translational Imaging Center, University of Southern California, Los Angeles California 90089, United States of America. [3]Rosalind Franklin Institute, Harwell Campus, Didcot OX11 0FA, United Kingdom. ✉e-mail: falkschn@usc.edu; marco.fritzsche@kennedy.ox.ac.uk

into clusters for productive signalling in B cells. Structural studies on TNF receptor superfamily member 1 (TNFRSF1) raise the possibility of an oligomerisation (dimerisation) of CD40 in a ligand independent manner[18,19].

Dynamic changes in molecular diffusion and oligomerisation can be measured by single particle tracking (SPT) approaches which traditionally rely on small concentrations and bright labels[20,21]. Though combination of photo-switching and single molecule localisation microscopy with SPT is paving new avenues for super-resolved dynamics. The exciting methods around MINFLUX provide unforeseen detail yet are still considered highly complex methodologies[22]. Another complementary approach to study diffusion dynamics is fluorescence recovery after photo-bleaching (FRAP), which does not require single molecule concentrations and assesses bulk properties of fluorescent tracers at longer length- and time-scales, but importantly yields information on a potential immobile fraction of molecules as well as reaction dynamics[23,24]. The fluorescence fluctuation spectroscopy (FFS) based approaches are attractive methods as they do not necessitate isolated single molecule events and provide sensitivity over a large range of concentrations[25,26]. In FFS, fluctuations caused by molecules entering and leaving the observation volume, defined by the microscope's point spread function (PSF), are analysed by, for instance, autocorrelation, revealing the underlying molecular dynamics of the intensity changes at this point in space. The most prevalent FFS based method is fluorescence correlation spectroscopy (FCS). Since its introduction in the 1970s, many derivative methodologies based on fluctuation analysis have been introduced to study dynamics, sub-diffusion, diffusion mode, or concentration and oligomerisation. These methodological advances continue to push sensitivity and the dynamic range for measuring molecular organisation and dynamics up to this day[27–30]. Examples include Number&Brightness (N&B) analysis[31], cumulant analysis[32], fluorescence intensity distribution analysis (FIDA)[33], and photon counting histogram (PCH)[34] which represent only a few attempts to provide a window to biomolecular organisation. N&B can be performed on a confocal laser scanning microscope or on a camera-based system. The acquisition and analysis of fluctuations in consecutive frames allow measurement of molecular brightness and concentration of molecules, but not diffusion. Cumulant analysis, FIDA, and PCH exploit mathematical analysis of the intensity distributions from single point data allowing higher temporal resolution but are limited in spatial coverage and statistics. Spatial sampling methods such as SpIDA (spatial intensity distribution analysis) or 2D-FIF (fluorescence intensity fluctuation spectrometry) overcome this issue, and are performed on whole images, but do not provide insights into diffusion dynamics[35–37]. Notably, 2D-FIF can be used to investigate fractions of oligomers and monomers[38–40]. Some image correlation methods allow for diffusion measurements such as raster image correlation spectroscopy (RICS) and spatio-temporal image correlation spectroscopy (STICS), but average over larger areas and lose sensitivity with regards to oligomerisation[41,42]. While fluctuation analysis methods are very powerful, other techniques have provided insights into oligomerisation and diffusion. SPT studies, for example, pioneered by Kusumi's laboratory[43,44], single molecule methods[45,46], or combinations thereof like thinning out clusters while conserving stoichiometry of labelling (TOCCSL)[47] or single molecule photobleaching analysis[48] represent only a few examples. Instrumentation to reliably measure diffusion dynamics and oligomerisation with single-molecule sensitivity in living cells is not widely available or only accessible to expert users hampering our abilities to further derive insights into these dynamic processes. Moreover, most of the methods are limited by the diffraction of light. The introduction of stimulated emission depletion nanoscopy and its combination with FFS approaches, yielded promising insights into molecular sub-diffusion modalities, but relies on dedicated hardware and software tools with specialised fluorescent dyes limiting their widespread application[49,50].

In summary, the existing methodologies often require highly specific, non-standard analysis tools for the quantification of biomolecule organisation and/or diffusion dynamics, hampering progress and broad availability of systematic biomolecule organisation analysis.

Exploiting the statistics from spatially resolved fluctuation spectroscopy acquisitions can give additional insights into sub-diffusion dynamics and potentially oligomeric states without the need for specialised equipment[3,51–53]. In scanning fluorescence correlation spectroscopy (sFCS) the observation volume is rapidly scanned along a line or circle yielding a set of dynamic measurements across space[54,55]. Consequently, the sFCS data contain a variety of statistical information on concentration, diffusion, brightness and thus oligomerisation. However, existing acquisition and analysis pipelines have not systematically exploited this full wealth of available information and therefore quantification of biomolecule diffusion and oligomerisation simultaneously has previously been challenging in living cells.

Here, we present the brightness-transit statistics (BTS) method enabling the simultaneous quantification of the diffusion and oligomerisation dynamics of biomolecules from sFCS measurements. Two-dimensional (2D) BTS histograms allow intuitive visual presentation of the diffusion times and oligomerisation (based on molecular brightness) at a population level, across multiple measurements and points in space. We initially validate the approach by means of computer-simulations and by experiments in vitro using induced oligomerisation of EGFP-His on reconstituted supported lipid bilayers (SLBs). Then we exploit the approach to investigate the oligomerisation and dynamics of CD40 on SLBs with and without interaction partners. Analysis with the BTS method thus enables systematic straightforward quantification of biomolecule organisation in vitro and in living cells.

## Results

We introduce the BTS method combining transit time (average time of molecules crossing the confocal detection volume in seconds (s)) statistics and brightness (counts per molecule; cpm in counts per second (Hz)) statistics (Fig. 1a, b). The BTS data are acquired in the same way as a line scan (xt) fluorescence fluctuation experiment wherein the same line of pixels is scanned $10^5$ times. This results in an array of intensity fluctuations across space which is further auto-correlated to yield correlation curves for every pixel. The transit times are then obtained from fitting the sFCS autocorrelation curves with a 2D diffusion model and the brightness is obtained from multiplying the average intensity (first moment) per pixel with the respective correlation amplitude (G(0) from fitting) at that location[31]. The detailed BTS workflow along with some tips to optimise the measurements is presented in Supplementary Fig. 1. The BTS method focusses on the population-level statistics from acquisitions over multiple areas by pooling and histogramming their fitted data and then comparing histograms obtained from different conditions. Thus, we disregard the spatial information from each individual scan. Brightness can also be calculated using the first and second moment of the intensity distribution only (no correlation), this has, nevertheless, proven impractical due to the need for daily detector calibrations. We employ this analysis on SLBs with and without interaction with immune cells to investigate changes in dynamic organisation, i.e., diffusion coefficient, oligomerisation and diffusion mode (see cartoon in Fig. 1a). This makes the BTS method suitable for the quantification of many biological diffusion processes at the membrane[53,55], compared to other methodologies (Supplementary Table 1). The nature of the simultaneous acquisition of both transit time statistics and brightness statistics allows then for the computation of 2D histograms correlating transit time and brightness statistics in an intuitive way (see Methods), to which we will refer in the following to as 2D BTS diagrams. Shifts of the peak position in the BTS histograms in the direction of the diffusion or brightness axis, respectively, indicate quantitative changes of

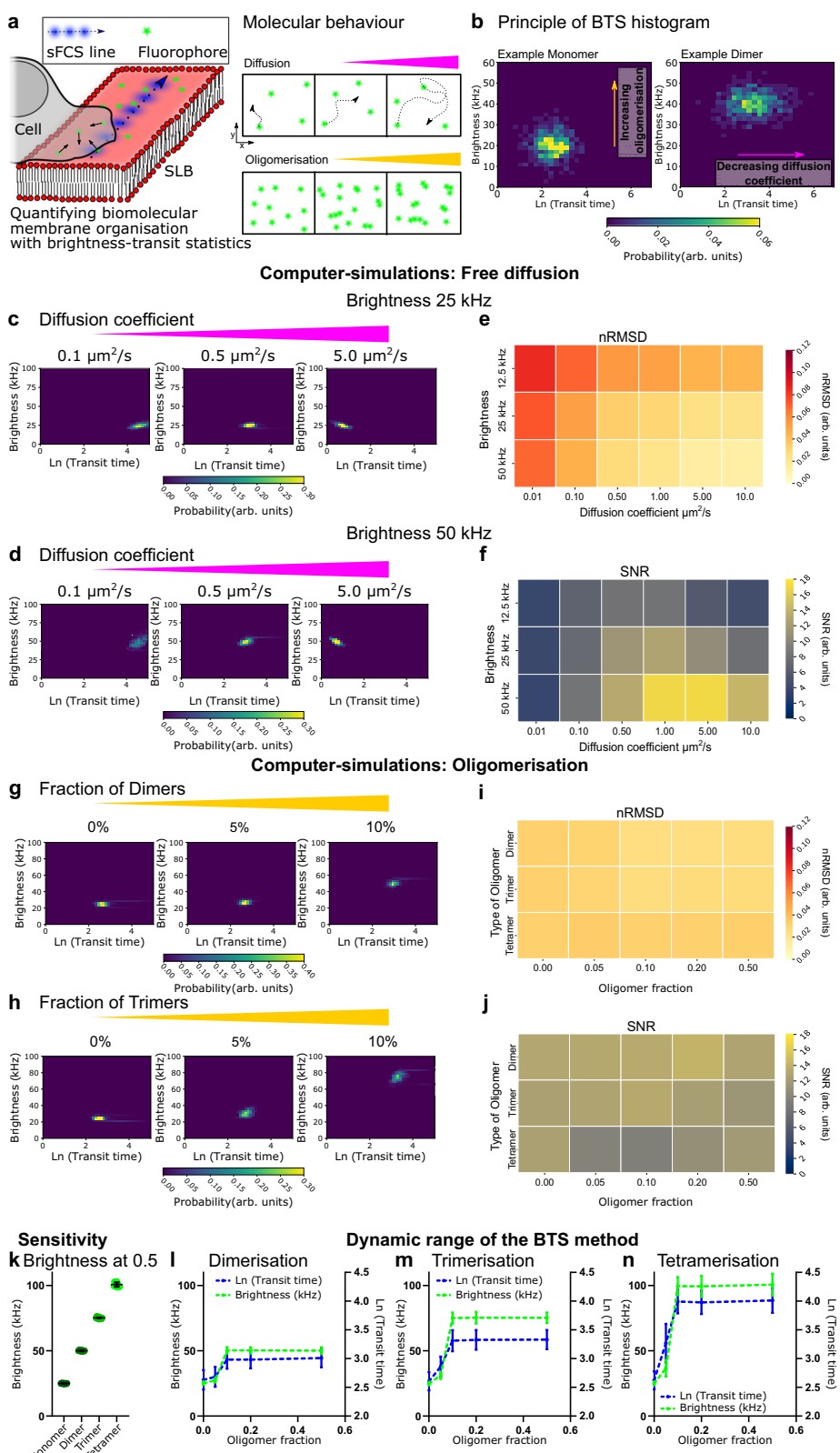

the diffusion coefficient or oligomerisation and thus organisation of the sampled ensemble of molecules (Fig. 1b). Note, we present the natural logarithm of the transit time (Ln (transit time (ms)/(ms))) as a convenient dimensionless measure which is normally distributed for freely diffusing molecules[53]. The transit times themselves are expressed in units of milliseconds (transit time (ms)).

## Computer simulations

We first set out to establish the working principle of the BTS method in two sets of computer-simulations (see Methods). With the first simulated data, we investigated the capacity of the BTS method to recover the diffusion parameters assumed for the simulations. For this, we simulated a large ensemble of freely diffusing particles assuming

**Fig. 1 | BTS diagram to investigate oligomerisation and diffusion dynamics.**
**a** Cartoon sketching the sFCS acquisition in the plane of an SLB (left) and changes in biomolecular organisation (right). Increase in diffusion coefficient (magenta) causes faster molecular movement, increase in oligomerisation causes higher molecular brightness. **b** Hypothetical BTS diagram demonstrating fast monomeric diffusion (left) versus slowed-down oligomeric diffusion (right). **c, d** Computer-simulations of freely diffusing molecules sampled by sFCS. Change in brightness (c to d) shifts the distributions on the y-axis and changes in diffusion coefficient (as indicated on panels) shifts distribution on x-axes. 10 measurements yielding 500 curves per condition were simulated. **e, f** Assessment of sFCS raw data quality for the simulations of free diffusion across the parameter space. Better data is represented by a low nRMSD value or a high SNR value, respectively. **g, h** Computer-simulations of binary mixtures (monomers and dimers g and monomers and trimers h) with increasing fraction of molecules in an oligomerised state (as indicated). Number of molecules in the simulation was kept constant and 500 curves

were simulated per condition. **i,j** sFCS raw data quality for the simulations of oligmerisation given by nRMSD and SNR evaluation. **k** Average brightness from the simulations on oligomerisation (monomers plus either dimers, trimers or tetramers) as described before (fraction 0.5, solid lines represent averages and standard deviations of the mean values from 10 simulations with 500 autocorrelation curves total). For the four sets of simulations the brightness values were significantly different from each other with $p < 0.01$ (one-way ANOVA with Tukey's test for multiple comparisons). **l–n** Changes in average transit time and brightness with changing fraction of oligomerisation for monomers plus dimers, trimers or tetramers, respectively. Dots represent averages and error bars standard deviations from 10 simulations including 500 curves. The brightness and transit time values for all fractions ≥0.1 were statistically significantly different from the monomer condition (i.e., at oligomer fraction=0.0, $p < 0.01$; two-way ANOVA with Tukey's test for multiple comparisons).

different transit times ($\tau_D$) that span over three orders of magnitudes: $\tau_{D,1} = 1038.7$ ms, $\tau_{D,2} = 103.9$ ms, $\tau_{D,3} = 20.8$ ms $\tau_{D,4} = 10.4$ ms, $\tau_{D,5} = 2.1$ ms, $\tau_{D,6} = 1.0$ ms; corresponding to diffusion coefficients: $D_1 = 0.01$ $\mu m^2/s$, $D_2 = 0.1$ $\mu m^2/s$, $D_3 = 0.5$ $\mu m^2/s$, $D_4 = 1$ $\mu m^2/s$, $D_5 = 5$ $\mu m^2/s$, and $D_6 = 10$ $\mu m^2/s$ assuming a Full Width at Half Maximum (FWHM) of 240 nm and a Gaussian observation volume. We tested two different sFCS scanning sampling frequencies (1000 Hz and 2000 Hz), and at three different molecular brightness values: 12.5 kHz, 25 kHz, 50 kHz (Fig. 1c, d and Supplementary Fig. 1). To facilitate comparisons with the literature, we report the simulations against the diffusion coefficients. Transit times, logarithmic transit times, and corresponding diffusion coefficients are presented in Supplementary Table 2. The absolute value of the transit time is important when considering the sampling frequency for sFCS, which basically defines the temporal resolution of the experiment[53,55]. Consistent with our expectations, the qualitative shape of the BTS histograms and their characteristic peak positions reported robustly and reliably on the pre-defined transit times and thus diffusion coefficients for all given molecular brightness of 12.5 kHz (Supplementary Fig. 2a, b), 25 kHz (Fig. 1c and Supplementary Fig. 2a, b), and 50 kHz (Fig. 1d and Supplementary Fig. 2a, b). Notably, no qualitative changes were observed in the computer-simulations for the two different sampling frequencies of 1000 Hz and 2000 Hz (Supplementary Fig. 2a, b).

To determine the potential bias and dynamic range of the BTS method, we assessed the fluctuation data quality of the simulations using two independent quantitative metrics including the normalised root-mean-square displacement (nRMSD) of the autocorrelation fits[26] and the signal-to-noise ratio (SNR)[55,56] of the autocorrelation curves for the various transit times and brightness settings (see Methods, Fig. 1e, f and Supplementary Fig. 3). These nRMSD and SNR maps consistently revealed high confidence in the sFCS raw data and fits, BTS measurements, across the parameter space with the increased confidence for larger brightness for both scanning frequencies (Supplementary Fig. 3a, b). Visual inspection reveals that the BTS peak intensity seemingly degrades with lower SNR and higher nRMSD for slowly diffusing molecules. Strikingly, for most of the diffusion coefficients and brightness values tested, the BTS method succeeds to recover the input diffusion values (<5% error; Supplementary Fig. 4), highlighting the experimental power of sFCS and the BTS method for biomolecular diffusion quantification. Notably, larger sampling scanning frequencies provide overall improved SNR. We observed a minor quality decay at the boundaries of the tested parameter space (Supplementary Fig. 4). The evaluation with the nRMSD and SNR values in addition to just examining the BTS histograms is necessary as the histogram spread can be misleading. The brightness in the histograms is not normalised as we aim to present raw brightness values to enable comparison between datasets, dyes, and setups. This causes, for example, a seemingly larger spread in the BTS histograms in Fig. 1d versus 1c for 0.1 $\mu m^2/s$ but is just a result of binning over more values.

Even given the same standard deviation (for example 1 sigma), the spread of values for a large mean value will be wider in absolute numbers on a histogram (see Supplementary Fig. 5 for normalised histograms).

In the second type of computer-simulations, we investigated the capacity of the BTS approach to recover oligomerisation (see Methods). We simulated multiple conditions, in which monomers are mixed with either dimers, trimers, or tetramers at various oligomerisation fractions ($f_1 = 0$, $f_2 = 0.05$, $f_3 = 0.1$, $f_4 = 0.2$, and $f_5 = 0.5$; higher $f_i$ indicates more oligomer, e.g., at $f_5$ half of all diffusing molecules are oligomers while the number of particles in the simulation is constant; Fig. 1g, h and Supplementary Fig. 6). Diffusion coefficients of the corresponding oligomerisation fractions were adapted to account for reduced oligomer mobility (see Methods). Visual inspection and statistical comparison of the resulting BTS diagrams revealed that the presence of oligomers was immediately apparent even at low fractions of oligomerisation ($f_1$ to $f_3$: ~5–10% oligomerisation; see Fig. 1g, h and Supplementary Figs. 6–8). Interestingly, for all simulated conditions, the fluctuation data qualities as assessed using the nRMSD and SNR values are similar suggesting no obvious bias in the raw data due to the presence of the brighter oligomers (Fig. 1i, j). Consistently, all simulation conditions for dimerisation, trimerisation, or tetramerisation confirmed the same qualitative trend, highlighting a high sensitivity of the BTS method with regards to detecting oligomerisation at a fraction as low as $f_2 = 0.05$ (Fig. 1k–n; the plotted brightness values represent the mean value of all cpm from a set of simulations and Supplementary Fig. 8 shows statistical differences of the BTS distributions). Furthermore, already at a fraction of $f_5 = 0.5$ the detected average brightness only reflects the oligomers while omitting the monomers (Fig. 1k). Notably, the diffusion and the brightness show different sensitivities, for example, 5% of dimers in Fig. 1g shows barely a change in diffusion but a clear shift in brightness. Consequently, the BTS analysis is very good at detecting oligomerisation i.e., picking up on small fractions of oligomers but can conversely not differentiate the fraction of monomers and oligomers when they are homogeneously distributed in space. Notably, the computer-simulations explored transit times even beyond the reasonable dynamic range of sFCS experiments. Generally, sFCS sampling needs to be performed sufficiently fast to capture fast diffusion (> 10 $\mu m^2/s$ large diffusion coefficients) and sufficiently long to obtain slow diffusion (~0.01 $\mu m^2/s$ low diffusion coefficients)[55].

## Oligomerisation of EGFP-His
Having established the sensitivity and dynamic range of the BTS method computationally, we next confirmed the experimental power in vitro utilising His-tagged ($His_6$) enhanced green fluorescent protein (EGFP-His) tethered via a nickelylated lipid to an SLB (see Methods and cartoon in Fig. 2a). We employed a two-step strategy for the induction of the artificial oligomerisation of EGFP. We dimerised the EGFP protein with anti-GFP antibody (monoclonal mouse IgG) and further

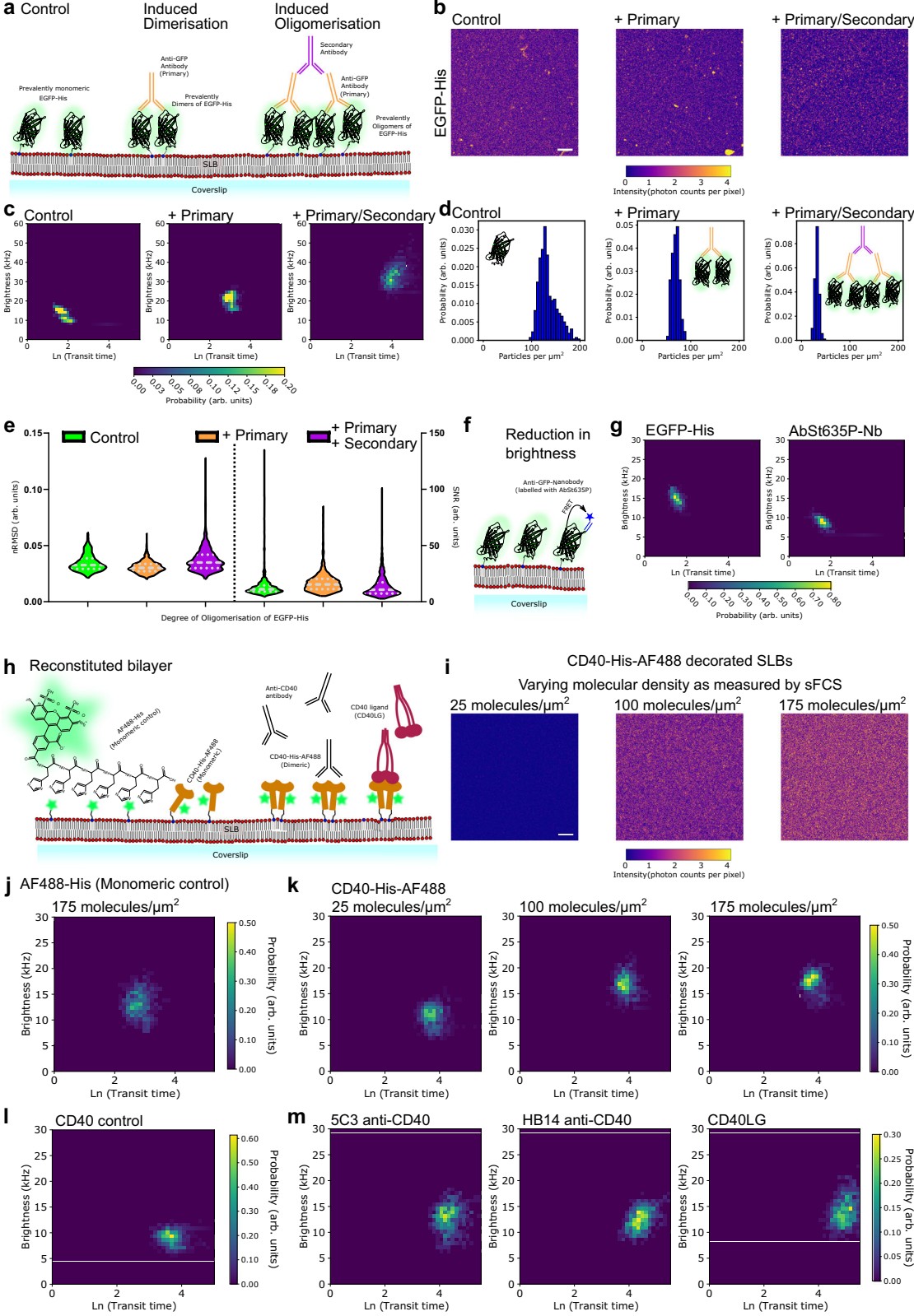

oligomerised the EGFP-antibody complex exploiting an anti-mouse antibody (goat anti-mouse IgG). Visual inspection reveals qualitative differences in the resulting sFCS raw data and slight changes to the distribution of fluorescence intensity in confocal microscopy images (Supplementary Figs. 9, 10, Fig. 2b). For the measurements, we avoided areas of inhomogeneous bilayers or large bright patches. Strikingly, in the homogeneous areas, the BTS histograms displayed pronounced

changes upon the presence of the primary and secondary antibodies (Fig. 2c). While both the control conditions with the EGFP alone and the EGFP with anti-GFP antibody displayed a distinct peak, the BTS diagram after addition secondary antibody revealed a broader distribution of values (possibly indicative of a mixture of oligomers). This was consistent with the BTS results observed in the computer-simulations for larger oligomers and slower diffusion dynamics

**Fig. 2 | Oligomerisation in vitro on reconstituted supported lipid bilayers.**
**a** Cartoon depicting the experimental set-up of a supported lipid bilayer with
tethered His-EGFP (visualised as green beta barrel inspired by PDB 1f0b[102]). Oligo-
merisation was induced by use of primary (orange) and secondary antibodies
(magenta). **b** Confocal images of SLBs composed of DOPC and DGS-Ni-NTA (4%)
labelled with EGFP-His (left), treated with primary (mouse) anti-GFP antibody
(middle), and addition of secondary antibody (anti-mouse, right). Scale bar is 5 μm.
Representative images from >3 independent experiments. **c** BTS diagram for EGFP
(left, control), EGFP and anti-GFP antibody (+Primary, middle), and EGFP, primary,
and secondary antibody (+Primary/Secondary, right). All BTS histograms are sig-
nificantly different from each other with $p < 0.01$ (permutation test). **d** Changes in
numbers of particles per area as determined by sFCS. **e** Experimental fluctuation
data quality measured by nRMSD and SNR as violin plots (dashed grey lines
represent median and dotted lines represent quartile values). **f** Cartoon depicting
the interaction of a membrane-anchored EGFP with a fluorescently tagged nano-
body undergoing energy transfer (FRET) **g** BTS histograms for EGFP (control, left)
and with anti-GFP nanobody (labelled with AbberiorSTAR635P, right), $p < 0.01$
(permutation test) **h** Cartoon of the reconstituted system for investigation of CD40

organisation. AlexaFluor488 conjugated to a His$_6$-tag (AF488-His) was used as a
monomeric control (left). Recombinant CD40-His was labelled with AF488 (orange
with green star). Perturbations performed with anti-CD40 antibodies (black) or
recombinant CD40LG (red). **i** Confocal images of SLBs with varying amounts
(measured by sFCS) of CD40-AF488-His. Scale bar is 5 μm. **j** BTS histogram for
AF488-His on a SLB as monomeric control **k** sFCS BTS diagrams for different sur-
face concentrations of CD40-AF488-His. 25 molecules/μm² is significantly different
from 100 molecules/μm² ($p < 0.01$) whereas 175 molecules/μm² is not significantly
different from 100 molecules/um² ($p > 0.05$, permutation test). **l, m** sFCS BTS dia-
gram for CD40-AF488-His (monomeric control, l) and under perturbation (m) with
anti-CD40 antibodies (5C3 and HB14) or recombinant CD40LG-His (all unlabelled).
Conditions in m are significantly different from the control in n with $p < 0.01$
(permutation test). 5C3 is not significantly different from HB14. CD40L vs 5C3
$p = 0.1$ and CD40L vs HB14 $p = 0.01$ (permutation test). The experimental histo-
grams are representative data from one repeat and integrate more than 10 sFCS
acquisitions on various bilayer locations (>500 autocorrelation curves) acquired at
ambient temperature (24 °C).

(Fig. 1c, d). Also, the histogram for the control condition let us
appreciate some heterogeneity in EGFP organisation on the bilayer
probably due to the fact that EGFP can dimerise by itself (Kd ~
0.11 mM)[57] as confirmed by experiments with different concentrations
of EGFP-His (Supplementary Fig. 11; we employed low concentrations
of EGFP-His to minimise this effect in the experiments on induced
oligomerisation). Furthermore, the addition of equimolar secondary
antibody is important as excess of secondary antibody can lead to the
formation of large, cross-linked, essentially immobile clusters that
cannot be analysed with a method relying on the motion of particles
(Supplementary Fig. 12). The reductions in the average number of
particles per micron-squared (from 134 molecules/μm² to 69 mole-
cules/μm² to 32 molecules/μm², respectively) inferred from the sFCS
acquisitions is suggestive of transitions from monomers to dimers and
dimers to tetramers. (Fig. 2d). Analogous to the simulations, we
checked the fluctuation data quality using the nRMSD and SNR metrics
observing similar values as for the simulated data with only a slight
degradation of the quality for the oligomeric state probably due to
slower diffusion (Fig. 2e). The in vitro BTS data on EGFP-His are sum-
marised in Supplementary Table 3.

We then aimed to test whether quantitative changes in brightness
independent of diffusion could be detected by the BTS histogram
analysis. One possible scenario to decrease brightness of fluorescently
labelled biomolecules is through fluorescence resonance energy
transfer (FRET). To test the performance of the BTS method, we
treated the EGFP-His on the SLBs with FluoTagQ nanobody, a small
GFP binding protein, labelled with AbberiorSTAR635P[58]. Consistent
with our expectations, the BTS histograms displayed a reduced
brightness but no significant changes in the transit time (Fig. 2f, g and
Supplementary Fig. 13). Notably, the combination of brightness and
transit time analysis as offered by the BTS approach revealed these
changes but they were invisible to the traditional transit time analysis
(Supplementary Table 3) as the diffusion coefficient is governed by the
membrane proportion of the diffusing particle (changing the soluble
part of the protein does not contribute to the diffusion coefficient as
much as much as changing the area embedded into the membrane
according to the Saffmann-Delbrueck-model[59,60]).

**Diffusion and oligomerization of CD40**
Having established the experimental power of the BTS method in vitro,
we next aimed to demonstrate its application with biomolecules
labelled with organic fluorescent dye on the surface of the SLB
exposed to recombinant ligand and finally to ligand on living cells. We
chose to quantify the dynamics of the lymphocyte receptor CD40 and
its interaction with homotrimers of its ligand CD40LG. It has been
suggested that CD40 binds at the interface of two of the three

monomers of CD40LG implying the possibility that in such symme-
trical configuration, and depending on the molecular densities of
CD40, CD40LG trimers might promote CD40 oligomerisation at the
IS[14,15]. Following the same strategy as EGFP-His, we decorated bilayers
with His-tagged CD40 ectodomain labelled with AlexaFluor488 and
used a His-tagged AlexaFluor488 as monomeric control (see cartoon
of experimental setup in Fig. 2h, specifically CD40-CA-His$_{12}$ labelled
with AlexaFluor488 and His$_6$-AlexaFluor488 conjugate, also refer to
Supplementary Fig. 14 for AF488-His raw data). Additionally, the SLBs
were blocked with Bovine Serum Albumin (BSA) to prevent non-
specific interactions and to allow experiments using living cells that
might otherwise non-specifically bind and activate on the SLBs[61]. The
BSA blocking caused a noticeable slow-down of the diffusion (longer
transit times) and an increase of the brightness of AF488-His on the
SLB (Supplementary Fig. 15). Therefore, we point out that performing
the control experiment (i.e. monomeric control) under the same
conditions as other experiments is paramount. We first titrated CD40-
His monomers fluorescently tagged with AlexaFluor488 (AF488,
labelled at fluorochromes per protein (F/P) < 0.7 meaning that most
proteins carry one or no label) on the SLB (Fig. 2i). Consistent with our
expectations, we observed increased intensity in confocal fluores-
cence images (Fig. 2i) but unexpected changes in the corresponding
BTS diagrams (Fig. 2j, k; raw data presented in Supplementary Fig. 16.
The BTS diagrams revealed pronounced quantitative changes in
molecular diffusion dynamics and oligomerisation of CD40 as com-
pared to the diffusion and brightness of monomeric AF488-His control
upon change in concentration on the SLB (Fig. 2j, k), suggesting that
CD40 has the capability to spontaneously form at least dimers as a
function of molecular density (as calculated from the number of par-
ticles in sFCS). For the deliberate induction of dimers, we used anti-
CD40 antibodies (in this case two agonistic antibody clones, 5C3 and
HB14, which bind two different portions of the CD40 molecule)[16].
These experiments displayed similar or slightly reduced changes
in transit time and brightness on the SLB at low concentration
(25 molecules/μm²) as compared to the high concentration of CD40
(Fig. 2l, m), highlighting that the binding strength or affinity of the
antibodies were smaller than for the interaction between the EGFP-His
and the respective antibody. Next, we used soluble recombinant
CD40LG (His-tagged but unlabelled) to mimic the native ligand
binding[62]. Visual inspection displayed a pronounced reduction in dif-
fusion coefficient in the BTS histograms with a small fraction of
molecules exhibiting a higher brightness as compared to the anti-
bodies (indicating potential for oligomerisation). Nevertheless, the
increased BTS peak position, in brightness and transit time, was clear
as compared to the monomer control (Fig. 2l). Together, these
experiments demonstrated the potential for CD40 oligomerisation

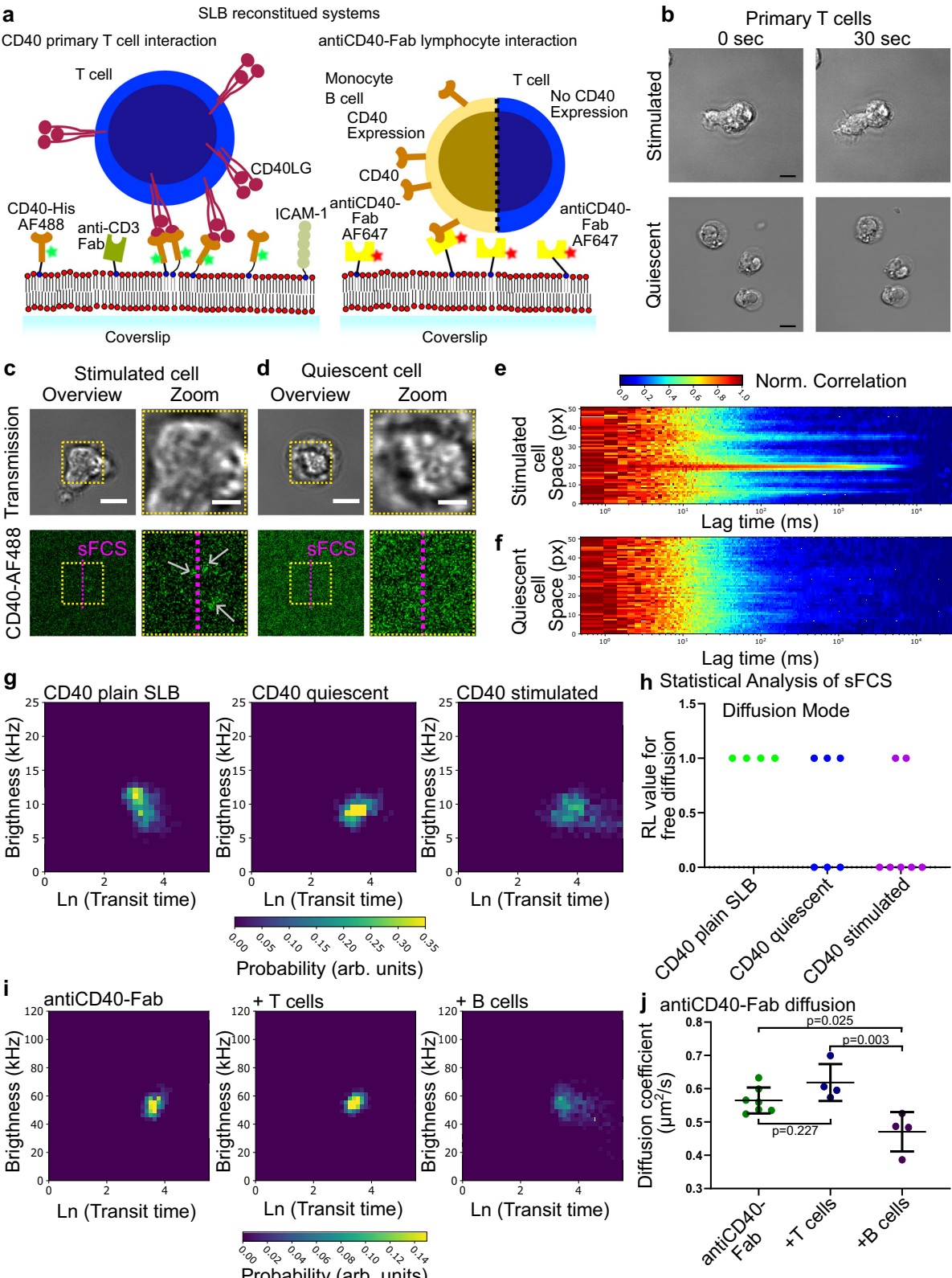

either spontaneously induced by (local) high concentration or in presence of CD40LG (cognate ligand) as well as anti-CD40 antibodies (artificial oligomerisation).

Next, we repeated the experiments in the presence of primary CD4[+] T-cells from healthy donors (see Methods), which were either stimulated (CD40LG[high], expressing high levels of CD40LG) or left untouched (quiescent) with low to null surface expression levels of

CD40LG, respectively[63] (schematic illustration in Fig. 3a). To facilitate cell bilayer interaction and ensure activation, the SLB was also decorated with ICAM-1 and anti-CD3ε Fabs in addition to CD40 and blocked with BSA to prevent non-specific interactions as mentioned previously. For quiescent and stimulated primary CD4[+] T-cells we observed two clearly distinct phenotypes in transmission microscopy right after exposing the cells to the SLB (Fig. 3b). While the stimulated CD4[+]

**Fig. 3 | CD40 organisation. a** Cartoon of reconstituted SLBs. Left: Activated T-cell (blue) expresses CD40LG and interacts with the proteins on the SLB. Experimental molecular densities were: CD40-His-AF488 35-50, ICAM-1-His 200, and anti-CD3ε-Fab-His 30 molecules/μm². Right: AlexaFluor647 (AF647) labelled antiCD40-Fab-His tethered to SLB. Primary lymphocytes expressing CD40 (B cells and monocytes) or lymphocytes lacking expression of CD40 (CD8⁺ T cells) were incubated with the bilayer allowing to follow CD40 organisation on cells. **b** Brightfield channel from confocal time-series for a stimulated (top) and quiescent (bottom) primary T-cell shortly after addition to SLB containing CD40-His-AF488. Scale bars are 5 μm. **c, d** Confocal and transmission images of a stimulated T cell (c) and quiescent T cell (d) on SLBs (labelled with CD40-AF488-His). Overview scale bar is 5 μm and zoom scale bar is 1 μm. Grey arrows indicate bright clusters of CD40-AF488-His. All confocal images are representative of at least 3 independent repeats. **e, f** sFCS raw correlation carpets for primary T-cells on locations exemplified by the purple lines in (**c, d**). **g** BTS histogram for CD40-His-AF488 on SLBs without T-cells (left), with quiescent cells (middle) and with stimulated T-cells (right). Control and quiescent

and quiescent and stimulated are not significantly different ($p = 0.15$, $p = 0.99$ respectively; permutation test), control and stimulated are significant ($p < 0.01$). **h** Statistical analysis of sFCS transit time histograms indicating the relative likelihood (RL value) of free diffusion (value of 1 means free diffusion is the most likely cause of distribution shape). Every dot represents the assessment of one biological replicate ($n = 4$ for plain SLB, $n = 6$ for SLB with quiescent cells, and n = 7 for SLB with stimulated cells) for each pooling > 400 autocorrelation curves from various locations on >2 SLBs. **i** BTS diagram of antiCD40-Fab-His-AF647 as control (plain bilayer, left), incubated with T cells (middle), and B cells (right). Shifts in the BTS histograms are not significant $p > 0.05$ (permutation test). **j** Diffusion coefficients extracted from sFCS measurements in i reported as average per condition (ie., every dot represents >400 individual FCS curves pooled for each replicate, different donors for conditions with cells; $n = 7$ for antiCD40-Fab, $n = 4$ for T cells, $n = 4$ for B cells). Horizontal lines are mean values and error bars are standard deviation. *P*-values calculated using one-way ANOVA with Tukey's test for multiple comparisons. All data were acquired at 37 °C.

T-cells were actively exploring the environment and were clearly changing shape over time, the quiescent CD4⁺ T cells displayed a passive phenotype with only minor morphological rearrangements such as membrane ruffling. Within a few minutes, stimulated CD4⁺ T-cells induced CD40 oligomerisation into clusters displaying bright fluorescent spots on the SLB (around 240 s after dropping the cells, Fig. 3c grey arrows, compare with quiescent cell in Fig. 3d), which was further confirmed by distinct peaks of slowed-down diffusion in the sFCS carpets (sFCS raw data, normalised auto-correlation across space, Fig. 3e). In contrast, quiescent CD4⁺ T-cells did not induce apparent CD40 reorganisation and the CD40 distribution was almost unperturbed (homogeneous SLB, Fig. 3d), which was again confirmed in the sFCS raw data (Fig. 3f and Supplementary Fig. 17). All sFCS data were acquired at the SLB-cell contact, scanning over the forming interface. Analysis with the BTS histograms, surprisingly, revealed that no mobile CD40 oligomers were present within the contact area formed by quiescent or stimulated primary CD4⁺ T-cells (Fig. 3g). The CD40 brightness was even slightly reduced while transit time increased when the cells were interacting with the SLB (Fig. 3g comparing the CD40 control on the bilayer without any cells present (left) to the conditions with cells (centre and right)), suggesting a shift towards monomeric form of CD40 with reduction in diffusion dynamics. The BTS data for CD40-His on the SLB in the presence and absence of the cells are summarised in Supplementary Tables 4, 5, respectively. The p-values from pairwise statistical comparison of the BTS histograms are given in Supplementary Table 6.

In light of these results, one would expect CD40 binding to CD40LG on the surface of the stimulated CD4⁺ T-cells, which may cause short, nano-scale, interactions and hindrances. We previously demonstrated how to differentiate free diffusion from hindered diffusion behaviour employing large sFCS transit time datasets using a statistical analysis pipeline[53]. Hindered diffusion transit time histogram data require a two-component lognormal model instead of a one-component lognormal model as caused by free diffusion to describe the sFCS transit time histograms. This can be unbiasedly evaluated using a model selection based on maximum likelihood estimations (Supplementary Fig. 18 and Methods). Applying this pipeline to the CD40 datasets for the control, the quiescent and the stimulated CD4⁺ primary T-cells allowed us to calculate relative likelihood values (RL values) for the different models and thus to determine whether CD40 molecules are diffusing freely or undergo hindrances (ie. transient binding). The diffusion behaviour of CD40 on the plain SLBs (in the absence of cells but in the presence of all other molecular components) was free with an RL value of 1 for free diffusion in all cases (Fig. 3h). In contrast, in 50% of the experiments with quiescent CD40⁺ T-cells and more than 70% of the experiments with activated/blasted CD4⁺ T cells, we find a qualitative shift from free diffusion to hindered

diffusion modes of CD40 on the SLBs indicating the presence of nano-scale binding events (see RL-values in Supplementary Table 7).

Finally, having quantified CD40 dynamics in vitro in the SLB in the presence of CD4⁺ primary T cells expressing CD40LG, we aimed to probe CD40 organisation in the plasma membrane of B lymphocytes using the BTS platform. For this, we chose to fluorescently label the CD40 with His-tagged anti-CD40 fab in the SLB as a proxy for the CD40 dynamics in the plasma membrane of the B cells. To ensure the ability to differentiate CD40 monomers from oligomers, we initially measured anti-CD40 fab tethered to the SLB in the absence of B cells to determine its monomeric characteristics (Fig. 3i left and Fig. 3a right for a cartoon of experimental setup). Next, we measured the same bilayer in the presence of B cells. We observed a notable slow-down of the fab but consistent with previous data no oligomerisation that would suggest induction of mobile cell surface CD40 oligomers (Fig. 3i, j). To ensure that those effects were indeed B cell specific, we also performed experiments with primary CD8⁺ T cells which do not express CD40. Notably, the interaction of T cells with the SLB did not change the diffusion coefficient as compared to plain bilayer controls. In addition, we found only monomeric CD40 and a reduced average diffusion coefficient in the presence of monocytes expressing CD40LG supporting the data on B cells (Supplementary Fig. 19). CD40 showed a shift from free diffusion to hindered diffusion on the bilayer when incubated with CD4⁺ T cells (see above, Fig. 3h). Analogously, the diffusion of the antiCD40-fab on the SLB shifts from free diffusion for the monomeric control and in presence of T cells to more hindered diffusion in presence of B cells (Supplementary Fig. 19). Altogether, analysis with the brightness-transit statistics method thus enables straightforward quantification of biomolecule organisation in vitro and with living cells.

## Discussion

We present a robust and straightforward pipeline to quantify biomolecular organisation in membranes. The BTS methodology enables measurements of oligomerisation and diffusion dynamics in the membrane in vitro and in the presence of living cells such as CD4⁺ T and B cells. The acquisition is based on a typical sFCS acquisition, xt-scan, which is easy to implement on any commercial confocal laser scanning microscope. We chose sFCS measurements to develop the brightness-transit statistics methodology as this technique gives insights into diffusion dynamics and brightness unlike other dynamic techniques such as FRAP or SPT which have their strength in elucidating biomolecular diffusion, directed motion, as well as reaction dynamics[64]. We determined high sensitivity to the presence of even small fractions of oligomers which in other methods with lower sensitivity to oligomers could yield biased results. While BTS is sensitive to detecting the presence of oligomers, estimating fraction of monomers

and oligomers in a mixture is not currently possible. Some spatial correlation methods offer a remedy but lack insights on diffusion dynamics[35,37]. An overview of the different techniques to study diffusion dynamics and/or oligomerisation is presented in Supplementary Table 1. The BTS approach is among the few that allows the study of both diffusion and oligomerisation, is possible using turn-key instruments, and provides intuitive presentation of the results. As a fluctuation-based technique the reason for this sensitivity for brighter particles is that the brightness determines the SNR[26,65–67]. One promising remedy to overcome this fluctuation bias could be the Bayesian statistical analysis to the brightness-transit pipeline although measurements at high concentrations may pose a major challenge[68]. Nevertheless, for the BTS approach (as for any fluorescence fluctuation-based analysis) the acquisition time and the sampling frequency are important factors for data quality[53,55]. In addition, photo-stability or conversely photo-bleaching of fluorophores need to be considered as the analysis relies on their constant average intensity. For all data acquired with AF488, we did not need to apply photo-bleaching corrections as we observed no systematic decrease in intensity (see Supplementary Figs. 9, 14, 16, 17). Notably, due to the rapid movement of the beam, the sFCS approach induces less photo-bleaching as compared to point FCS at the same laser power[69,70]. While photo-bleaching can be corrected for, it should be avoided if possible. Furthermore, care should be taken to prevent any photo-physical effects such as optical saturation that can bias the analysis[71]. We acquired point and scanning fluctuation data within the linear regime where average intensity scales with excitation power. Experiments also come with noise inherent to the experimentation, sample movements, drift or electronical noise degrade the SNR as compared to simulated data.

The computer-simulations show the sensitivity of fluctuation-based approaches towards the presence of even a small fraction (~5%) of oligomerised molecules. Together with the careful experimental testing using the EGFP-His antibody system, we could determine sensitivity and applicability of the BTS approach covering a range of diffusion coefficients from 0.1 $\mu m^2$/s to 10.0 $\mu m^2$/s (using simulations, Supplementary Table 2) and 0.1 $\mu m^2$/s to 3.4 $\mu m^2$/s (in experiments, Supplementary Tables 3–5). Thus, it covers dynamic biological processes from low diffusion coefficients of 0.001 – 0.05 $\mu m^2$/s (protein clusters such as G protein-coupled receptors[72], the T cell receptor[73], or the epidermal growth factor receptor[74]) over 0.1–2 $\mu m^2$/s (single pass transmembrane receptors such as CD4/CD8[75] or membrane anchored proteins like Lck or GPI-anchored proteins[76]) to faster diffusion coefficients >2 $\mu m^2$/s (of lipids and cholesterol[77,78]). Diffusion in the cyto- or nucleoplasm can be substantially faster but we are limiting the analysis to membrane bound molecules, and oligomerisation processes, which usually scale from 2–100 s molecules forming macromolecular complexes, clusters[2]. Changes in transit times and diffusion coefficients follow the change in molecular brightness indicating a potential bias towards sampling the brighter molecules. This is in accordance with previous findings that the SNR in FCS is ultimately determined by the cpm[26,65–67]. The BTS histograms in Fig. 2c reveal the expected changes in oligomeric state of EGFP but also display a larger heterogeneity as compared to the simulated data (Fig. 1g, h). It is inherent to an experimental system that multiple factors contribute to the dynamic state. While we focus on the induced change through the addition of the antibody, also other factors such as imperfections in the bilayer, misfolded (i.e., non-fluorescent) EGFP, multiple nickelylated lipids bound to EGFP, self-quenching of the fluorescent protein, or differential photo-bleaching kinetics between mono- and oligomer contribute to the BTS distribution. An experimental system is not "clean" as computer-simulations, for example, we avoided bright areas in the bilayer (that were otherwise homogeneous) and were mindful of the concentration to avoid self-dimerization of EGFP (Supplementary Fig. 11). While a photo-stable and purely monomeric His-tagged

fluorescent protein could be used, it is unfortunately not commercially available at this point and would also not mitigate the above-mentioned issues about misfolded proteins or self-quenching. Nevertheless, as we can clearly resolve the changes upon antibody addition, we believe these experiments serve well in demonstrating the BTS analysis. Yet, this highlights again the challenge of extracting exact fractions of molecules with very different molecular brightness in the populations of sampled molecules using fluctuation-based analysis[79]. Using the large statistics from sFCS or even imaging FCS acquisitions may present a remedy provided that the species can be separated in space[52,80–82]. As for the EGFP experiments and for the CD40 experiments, having a good monomeric control is crucial. The analysis relies on the assumption that the majority of EGFP-His (at the employed low concentrations) and the AF488-His are monomeric by themselves and also that the majority of the fluorophores are fluorescent; revelation of oligomerisation relies on that, but it is worth noting that especially fluorescent proteins can have a considerable fraction of dark molecules[83,84]. Therefore, carefully characterising the monomeric conditions is important and common to methods studying oligomerisation.

We provide insights into the CD40 diffusion and organisation in reconstituted systems and on living cells. We find that CD40 can oligomerise depending on concentration and in the presence of its ligand CD40LG as suggested by data from crystallography on TNF receptors where two different configurations of dimers have been indicated (parallel and anti-parallel)[18,19,85]. Ligand dependent and independent clustering could therefore potentially yield different types of dimers or oligomers. In fact, the BTS data on in vitro homo-oligomerisation induced by changes of concentration and the hetero-oligomerisation by antibodies or ligand hint slight differences (Fig. 2k vs Fig. 2m). The interpretation of the data in presence of ligand (or antibody) is more involved as those proteins may also interact with the SLB themselves (for instance via the His-tag of the recombinant CD40LG), or the BSA used to block the bilayer to prevent non-specific binding as it was the case for all presented experiments with quiescent and stimulated CD4$^+$ T cells[61]. It is also worth noting that the accuracy of brightness estimation from the BTS histograms is slightly affected by the decreasing diffusion coefficient (see free diffusion simulation in Supplementary Fig. 2).

Engagement of monomeric CD40 by quiescent and activated CD4$^+$ T cells has implications for how these cells sense antigen-presenting cells and how signals are processed. While CD40LG is believed to be a trimer, it has been proposed to be able to form a complex with α1β5 integrin and then interact with monomeric CD40[86] offering an explanation for the difference in CD40 oligomerisation when CD40LG is presented in solution versus on the surface of T cells. We observe the formation of larger clusters at the contact of the stimulated cells with the SLB. Nevertheless, most of these clusters were immobile on the scale of the fluctuation acquisition (~40 s) and do therefore not contribute to the fluctuation analysis. To improve cell-SLB contact and more closely mimic the physiological situation, we also reconstituted ICAM-1 and antiCD3ε-Fab. This is probably the reason why we also find quiescent cells spreading and showing partial introduction of hindrance in CD40 diffusion. In addition, a small proportion of the quiescent cells may express low amounts of CD40LG, which can interact with the CD40 on the SLB. Of course, we acknowledge that the SLB-cell interaction only represents an over-simplification of the cell-cell interaction case as, for instance, only a proportion of the molecules normally present on the cell surface were reconstituted using the His-tags and no extracellular matrix molecules or glycocalyx molecules were included. Only BSA/HSA has been used to block the bare SLB. A next generation system could consist of an SLB generated from cells expressing the proteins of interest[87]. Also, an intact cytoskeleton might make a difference to the organisation as cortical actin is

crucially involved in membrane organisation[88]. Additionally, the geometry and topography of the interaction could be reconstituted using large vesicles as artificial cells[89]. Furthermore, the biophysical properties of the interaction such as the stiffness of the substrate need to be considered and could be tuned, for example by using a gel supported bilayer[90]. Ultimately, of course, the CD40 and CD40LG organisation needs to be probed on B- and T cell simultaneously focussing on the full protein constructs at best in physiological context of blood or tissue[4]. As an intermediate step we used bilayer-bound anti-CD40 fab to capture the dynamics on the cells. This allows us to have an internal monomeric control (bilayer before addition of cells) but we cannot exclude that some fabs are not bound to CD40 on the cell. Alternatively, we could use CRISPR/Cas9 to genetically tag CD40 on the B-cells but that is currently challenging for primary cells. Furthermore, the timing of the synapse formation is likely very important as more and more CD40 is constantly being accumulated over time[91,92]. For the experiments we focussed only on the initiation of the signalling process but also later stages with secreted synaptic ectosomes which have high CD40LG densities would be interesting to study[91]. Yet, for the initiation of synapse formation and spreading we believe that a change from free diffusion to trapped diffusion of CD40 on the SLB rather than its oligomerisation is the driver of cellular signalling. It should also be considered that we could not investigate the data with the current framework for directed molecular flow which could be added to the workflow in the future. The true nature of biomolecular interaction is hard to address with a single fluorescent label and in addition the exact label to target ratio must be known to be able to interpret the results. Introduction of cross-correlation statistics and brightness (exploiting a second fluorophore) might pave the way to unveil more details and accurate stoichiometry. Yet, this would introduce more complexity to the measurements (chromatic aberration, matching of excitation beams, multi-colour correlations). The use of an additional reconstituted CD40 or CD40LG labelled with a different fluorophore could shed more light on the CD40 oligomerisation[79,84]. Conversely, any label or fluorescent tag can influence or biases molecular interactions. Novel tools based on interferometric scattering microscopy, enable mass photometry and recently allowed to study diffusion and oligomeric state without the need for labelling which also removes any possible artefacts from photo-bleaching[93]. While this is an incredible achievement, such measurements in the presence of a living cell seem still far beyond reach.

In conclusion, the brightness-transit analysis provides a broadly applicable methodology and straightforward data representation. Expanding the brightness-transit analysis further to quantification of large areas e.g. whole cell contacts might offer a perspective for a more complete characterisation of biomolecule interaction dynamics such as receptors, transcription factors, kinases, and their various ligands in the near future. Considering the broad applicability, straightforward implementation, usage of standard fluorescent proteins and dyes at turn-key microscopes, we envisage brightness-transit analysis to become an important tool for characterising biomolecule organisation in the plasma membrane of living cells.

## Methods

### Ethical statement
The conducted research complies with the relevant ethical regulations at the University of Oxford. Different T cell populations were isolated from leukapheresis reduction system (LRS) chambers from de-identified, non-clinical, and consenting healthy donors. The non-clinical issue division of the National Health Service and the Inter-Divisional Research Ethics Committee of the University of Oxford approved the use of LRS chambers (REC 11/H0711/7 and R51997/RE001).

### Cells and cell culture
Untouched T cells (CD4[+] and CD8[+] as indicated in figure legends), B cells, and monocytes were isolated from LRS chambers using immunodensity negative selection (RosetteSep™, StemCell™ Technologies). Isolated cells were then cultured at $2 \times 10^6$ cells/mL in RPMI 1640 media containing 10% of heat-inactivated foetal bovine serum, and 100 μM non-essential amino acids, 10 mM HEPES, 2 mM L-glutamine, 1 mM sodium pyruvate, 100 U/ml of penicillin and 100 μg/mL of streptomycin. In some experiments CD4[+] T cells were either kept quiescent or activated by blasting with Human T-activator CD3/CD28 Dynabeads (ThermoFisher) for three days in the presence of 100 U/mL of recombinant human IL-2 (Peprotech). After removal of activating dynabeads, the activated state was maintained by replenishing fresh media containing 100 U/mL of recombinant human IL-2 every 48 h. Expression of CD40LG was checked by flow cytometry before performing any dynamic experiments.

### Generation of Alexa Fluor 488-conjugated human CD40-CA-His$_{12}$
The ectodomain of human CD40 comprising amino acid positions 21 to 193 and including a C-terminal free cysteine and a 12-histidine tag was labelled using 1.2 molar excess of Alexa Fluor 488 C5 Maleimide (Thermo Fisher Scientific, #A10254) in the presence of a 10 molar excess of TCEP (Thermo Fisher Scientific, Pierce, #20490) for 2 h at RT in the dark. To reduce both the likelihood of proteins self-aggregating due to the formation of spontaneous linker S-S bonds and conjugating to an excessive number of dyes per protein, unlabelled proteins were kept in an environment with 5-fold molar excess of TCEP. After labelling, excess unconjugated dye was removed by passing the labelled protein solution through a 7,000 MWCO desalting column four times (Thermo Fisher Scientific, #89882) and subsequent overnight dialysis against 10 volumes of sterile PBS at pH 7.4. The Alexa Fluor 488 (AF488) conjugated CD40 was then centrifuged for 4 h at 120,000 x $g$ at 4 °C to remove large molecular weight aggregates. The efficiency of labelling expressed as fluorochromes per protein (F/P), was then estimated using spectrophotometry as indicated by the manufacturer. The CD40 ectodomain sequence used in this study is as follows:

ETGEPPTACREKQYLINSQCCSLCQPGQKLVSDCTEFTETECLPCGE
SEFLDTWNRETHCHQHKYCDPNLGLRVQQKGTSETDTICTCEEGWHCT
SEACESCVLHRSCSPGFGVKQIATGVSDTICEPCPVGFFSNVSSAFEKCHP
WTSCETKDLVVQQAGTNKTDVVCGPQDRLRCAHHHHHHHHHHHH.

### Labelling of antiCD40-His-AF647
The recombinant anti-CD40 fab was custom produced based on the anti-CD40 antibody by Absolute Antibodies (#Ab00129-10 6, clone G28.5). Stock concentration was 1 mg/mL. Labelling was performed analogously to labelling recombinant CD40. Briefly, the fab was labelled in 5 molar excess of Alexa Fluor 647 succinimidyl ester (#A20006, Thermo Fisher Scientific) at 4 °C over night. After conjugation free dye was removed using 7000 Da MWCO desalting column (Zeba, #89882, Thermo Fisher).

### Preparation of supported lipid bilayers (SLBs)
Supported lipid bilayers for experiments on CD40-His were prepared by vesicle fusion as described elsewhere[63,91]. A vesicle mix containing DOPC (Avanti Polar Lipids) and Ni-NTA-DGS (Avanti Polar Lipids) vesicles in water was used to yield a final composition of 12.5 mol% Ni-NTA-DGS on the bilayer. The solution was incubated on a 6-channel #1.5 glass cover slip slide (IBIDI, μ-Slide VI 0.5) for 30 min and was subsequently washed three times with HBS (pH 7.4) containing 1% HSA (Sigma Aldrich). Next, the SLB was blocked with 5% BSA (Sigma Aldrich) in HBS containing 100 μM NiSO$_4$ (Sigma Aldrich) for 20 min. After another round of washing, a solution of AlexaFluor488-labelled

CD40-His (specifically the extracellular domain of CD40-Cys-Ala-His$_{12}$ produced in house as described above) was added to the bilayer to yield molecular densities between 20 and 500 molecules/μm$^2$ as indicated. Fluorochrome to protein ratio (also known as F/P was determined using a spectrophotometer to be 0.7). The CD40-His density for all experiments with cells was between 20 and 50 molecules/μm$^2$ as measured by FCS during the acquisitions. For experiments with cells the CD40-His solution additionally contained ICAM-1/anti-CD3ε monomeric Fab clone UCHT1 (at final densities of 200 and 30 molecules/μm$^2$). Measurements were performed after three more washes with HBS/HSA. As a monomeric control CD40-His was replaced by AlexaFluor488-His (custom synthesis by Cambridge BioScience and kind gift by Prof. Erdinc Sezgin)[94]. Note that the SLBs for all experiments involving CD40 were blocked with BSA and HSA as described above to be consistent with previously acquired data and to prevent non-specific interaction possibly causing activation[61,63].

The supported lipid bilayers for the experiments on EGFP-His (OriGene CAT#: TP790050; the protein carries an N-terminal His$_6$ tag) were prepared using spin coating of a lipid mix containing 4 mol% Ni-NTA-DGS in DOPC at 1 mg/mL total lipid concentration (in CHCl$_3$:Methanol 3:1) as described in ref. 89. Briefly, the resulting lipid film after spin coating of 25 μL of the lipid mix onto 25 mm piranha edged glass cover slips for 45 s at 3200 rpm was hydrated and washed with HBS. After SLB formation, EGFP-His was added to a final molecular density of 100–200 molecules/μm$^2$ and incubated for 30 min before washing again with HBS.

Treatment with anti-CD40 antibodies (clones HB14 and 5C3 BioLegend #313002 and #334325) at 1 μg/mL, recombinant human CD40LG (TNFSF5) (carrier-free; BioLegend #591706), anti-GFP antibody (labelled with AlexaFluor647 (R&D systems, FAB42402R)), secondary antibody (anti-mouse, IgG labelled with Alexa Fluor 647, Invitrogen) and anti-GFP nanobody (FluotagQ labelled with Abberior-STAR635P, NanoTag) was performed after the system equilibrated. All in vitro experiments were performed at room temperature and all experiments with the primary T-cells were performed at 37 °C.

## Confocal microscopy
Confocal microscopy was performed on two different setups: Zeiss780 LSM and Zeiss980 LSM equipped with a 40 × 1.2 NA C-Apochromat W Corr FCS objective (Carl Zeiss). Before imaging, the correction collar was adjusted based on point FCS acquisitions (see section sFCS acquisitions). On the Zeiss780 LSM, EGFP and AF488 fluorescence was excited using the Argon laser line (488 nm) set to 0.3% or 0.5% (corresponding to 9 μW and 12 μW at the objective lens, respectively). Fluorescence was collected using the prism on ChannelS (GaAsP hybrid detector) between 500 nm and 600 nm. Excitation and emission were separated using the 488/594/633 MBS. The pinhole was set to 1 A.U. Typically, images were acquired in photon counting mode at a zoom factor of 5 resulting in an image size of 42.5 by 42.5 μm$^2$ (512 by 512 pixels). Transmitted light (brightfield) images were acquired simultaneously using the T-PMT. On the Zeiss980 LSM, AF488 and EGFP fluorescence was excited using 488 nm diode laser set 2% and 0.5% (corresponding to 48 μW and 12 μW at the objective lens, respectively). Fluorescence was collected using the prism on ChannelS (GaAsP - PMT detector) in the integration mode with gain set to 650 V and between 500 nm and 600 nm wavelength range. Images were acquired at a zoom factor of 16 resulting in an image size of 13.26 by 13.26 μm$^2$ (512 by 512 pixels). Interference reflection microscopy was performed on cells with laser power adjusted to <10 nW.

## Image processing
All image processing steps such as adjusting contrast/brightness, cropping for generating zoom-ins, performing temporal projection, or saving to.png-files were performed in FIJI[95].

## FLIM
The fluorescence lifetime imaging microscopy data were acquired on a Leica SP8 equipped with a fast lifetime contrast (FALCON) module (Leica Microsystems). EGFP fluorescence was excited using the white light laser and the 488 nm excitation band was selected. Fluorescence was collected through a HC PL APO 86 × 1.20 W motCORR STED WHITE water immersion objective in a window from 500 nm to 600 nm on an internal HyD-SMD detector. Laser repetition rate was set to 40 MHz to allow for a larger dynamic range for the lifetime determination. The time correlated single photon counting decays were analysed for the whole image within the LAS-X SMD routine. A bi-exponential model (tail-fit) was used for all data and the intensity weighted average lifetimes reported.

## sFCS acquisition
The sFCS data were acquired on the above described Zeiss780 LSM and Zeiss980 LSM set-ups. Both systems were calibrated in pFCS mode using the internal FCS routine and a solution of 50 nM AlexaFluor-488 or 10 nM Atto655 in water which have a known diffusion coefficients of 435 μm$^2$/s and 426 μm$^2$/s, respectively[96]. These were used to calculate the radius of the observation area ω which was subsequently used to calculate the area of the observation spot $A$:

$$D = \frac{\omega^2}{4\tau_D} \tag{1}$$

$$A = \pi\omega^2 \tag{2}$$

The transit time $\tau_D$ was obtained from fitting the autocorrelation curve (see below). The pinhole was aligned and the correction collar adjusted every day to yield maximum (and consistent) cpm. To acquire the fluctuation data the imaging mode was set to line and the line length set to 52 pixels at a zoom factor of 40 (resulting in pixel size of 100 nm/px) on Zeiss780 LSM and to 128 pixels at a zoom factor of 16 (resulting in pixel size of 104 nm/pixel) on Zeiss980 LSM. Highest sample scanning frequencies were chosen for each microscope. This corresponds to a pixel dwell time of 3.94 μs with a scanning frequency of 2069 Hz on the 780 system, and 0.98 μs with a scanning frequency of 3415 Hz on the 980 system. The same line was scanned between 100,000 and 200,000 times resulting in a total acquisition times of 48 to 60 s. It is worth noting that the scanning frequency (line time) and total acquisition time need to be optimised and matched to the diffusion dynamics under investigation (rule of thumb: the line time should be 5–10 times shorter than the transit time and the acquisition time should be about three orders of magnitude larger than the transit time to allow the correlation curve to converge). We refer the reader to published protocols for optimisation of acquisition parameters[50,97]. The data was saved as.lsm5-files (780) and czi-files (980). A typical measurement file (enabling to reuse the settings) is deposited along with the analysis code on GITHUB [https://github.com/Faldalf/sFCS_BTS].

The data on induced EGFP-His oligomerisation, on CD40-AF488 (in vitro and in presence of cells) were acquired on the Zeiss780 LSM. The titration experiments of EGFP-His and the experiments on the anti-CD40 fab were acquired on the Zeiss980 LSM.

## sFCS simulations
sFCS simulations were performed in Python based on previous implementations[53,55]. The code can be found on GITHUB [https://github.com/Faldalf/sFCS_BTS]. For free diffusion 400 - 1000 molecules were randomly initialised in a 5 by 8 μm$^2$ box and their movement based on a random walker model (with tuneable diffusion coefficient). The moving particles were sampled by a simulated sFCS acquisition with 50 pixels using a 240 nm FWHM Gaussian PSF. The acquisition

time was 30 s, pixel dwell time was 4 μs (for numerical reasons equal to the time steps of the simulation) and the sampling rate 1000 or 2000 Hz as indicated. Fluorescence was discretised using a Poisson distribution yielding photon counting data and allowing to tune the brightness of the molecules as in ref. 56. To speed-up simulations the calculations for multiple sets of particles were split up and performed on different CPUs using iPyParallel. Typically, 500 curves per condition were simulated.

For simulations on clustered/oligomerised diffusion a variable amount of brighter and slower particles was introduced to the simulation box (keeping the total number of molecules constant). To visualise the effect of oligomerisation on in the BTS histogram, we assumed that dimers, trimers and tetramers would show a reduction in diffusion coefficient by a factor 0.7, 0.5 and 0.25, respectively[59,60]. We assumed a linear increase in brightness with oligomeric state for the simulations.

## sFCS data analysis

The fluctuation data were correlated (sFCS) and fitted (sFCS, pFCS) using the FoCuS_point and FoCuS_scan software packages[55,98]. The first 10 s of each measurement were cropped off to remove fluorescence signal contributions from immobile molecules which would distort the temporal autocorrelation. Subsequent analysis was performed on the truncated data. The sFCS data on experiments with AF488 did not require spatial cropping or photo-bleaching correction. The data on AF647 (anti-CD40 fab) were corrected using a local averaging approach with a time interval of 16 s and were spatially cropped from 128 px to 72 px in FoCuS_scan. FCS data were fitted to an equation of the general form:

$$G(\tau) = G(0) \cdot (G_D(\tau) \cdot G_T(\tau)) + offset \qquad (3)$$

Where $\tau$ is the lag time, $G(\tau)$ the correlation, $offset$ displaying an offset from 1 or 0 and $G(0)$ referring to the amplitude (inversely proportional to the average number of molecules $N$ in focus). $G_D(\tau)$ refers to correlation contribution based on diffusive processes (ie. 2D or 3D diffusion) and $G_T(\tau)$ refers to correlation contribution based on photophysics such as triplet states. For fitting of pFCS acquisitions the offset was fixed to 1 and in sFCS acquisitions offset was varied. For the solution measurements (calibration with AF488) we used a 3D free diffusion model:

$$G_{3D}(\tau) = \left(1 + \left(\frac{\tau}{\tau_D}\right)\right)^{-1} \cdot \left(1 + \frac{\tau}{AR^2 \cdot \tau_D}\right)^{-\frac{1}{2}} \qquad (4)$$

Here, $\tau_D$ is the transit time and $AR$ referes to the aspect ratio of the observation volume which was varied between 1 and 10 and fixed during a second round of calibration fitting. The pFCS data showed a contribution of the fluorophore's triplet state which was fitted to the additional term:

$$G_T(\tau) = 1 + \left(\frac{T_T}{1 - T_T}\right) \exp\left(\frac{-\tau}{\tau_T}\right) \qquad (5)$$

Where $T_T$ is the triplet fraction and $\tau_T$ the characteristic triplet time (fixed to 5 μs).

For sFCS a simple 2D diffusion term was used as $G_D(\tau)$ for fitting the data:

$$G_{2D}(\tau) = \left(1 + \left(\frac{\tau}{\tau_D}\right)\right)^{-1} \qquad (6)$$

Fitting parameter were stored in Excel sheets and further processed in Python (3.7).

## BTS diagrams and analysis

Data analysis code can be found at GITHUB [https://github.com/Faldalf/sFCS_BTS]. The sFCS data were read into Python, using the pandas package and processed as pandas data frames. The transit time and brightness values were plotted as 2D histograms using matplotlib[99]. 2D bins were kept constant throughout a set of experiments for the bi-variate histograms comparing different conditions and histograms normalised using the pdf option. Reported values consist of averages and standard deviations calculated with the built-in functions in pandas.

## Calculation of nRMSD and SNR as sFCS raw data quality metrics

Quality of FCS fits and data were evaluated by normalised residuals mean squared displacement (nRMSD) and by signal-to-noise ratio (SNR), respectively, analogous as described in refs. 26,55,56. For the nRMSD we used the full lag time range from shortest lag time (based on scanning frequency, for 2000 Hz: 0.5 ms) to fitting maximum (10 s):

$$nRMSD = \frac{\sqrt{\sum_{i=1}^{N_i} (G(\tau_i) - fit(\tau_i))^2}}{G(0)_{fit}} \qquad (7)$$

Here $N_i$ denotes the number total lagtimes (not to be confused with the actual value of the lagtime), $G(\tau_i)$ the correlation value at the i-th correlation point $\tau_i$ and accordingly $fit(\tau_i)$ the value of the fit at the i-th correlation point. $G(0)_{fit}$ is the fitted correlation amplitude used for normalisation.

For the SNR we used the first 9 lag times ($N_k$) of an autocorrelation function and the variance ($var$) obtained over these $N_k$ from three chunks of the original curve (chunks had equal length (e.g. 10 s each for the simulations)):

$$SNR = \frac{1}{N_k} \sum_{i=1}^{i=N_k} \frac{G(\tau_i)}{\sqrt{var}} \qquad (8)$$

The fitting and data processing in this case was performed using custom code in Python (code available on GITHUB [https://github.com/Faldalf/sFCS_BTS]).

## Statistical analysis of sFCS data

The statistical analysis of sFCS transit time histograms for the determination of the diffusion mode (free diffusion versus nano-scale hindered diffusion, i.e. trapping) was performed as described previously[53]. The original analysis code (written in MATLAB) was re-implemented in Python and is available on GITHUB [https://github.com/Faldalf/sFCS_BTS]. This analysis pipeline works at the population level (pooled datasets from many sFCS acquisitions of multiple cells and locations). We pooled the data by experimental day so that one sFCS transit time histogram would be representative of multiple cells. Model selection for free diffusion (lognormal distribution) versus hindered diffusion (double lognormal distribution) was performed via a maximum likelihood estimation. From the optimised logarithmic likelihood vectors, we calculated the Bayesian information criterion and subsequently a relative likelihood value (RL value) for every tested model. The RL value is 1 for the model describing the data best and lower for all other models (we tested single lognormal, double lognormal and Gaussian models).

## Statistics

All in vitro experiments (EGFP-His and CD40-His oligomerisation experiments) were performed at least three independent times. For every day the data from multiple locations and SLBs were integrated. The experiments using the primary T and B cells as well as monocytes were performed in three batches of two independent donors each (six

donors in total). For every batch data from two experimental days (multiple bilayers and cells) were used for analysis. The presented BTS data in Fig. 3 are from one representative batch and the statistical analysis of sFCS data was integrated over all three batches and all donors (i.e., technical and biological replicates Fig. 3h).

BTS histograms and their statistical differences were compared pairwise as appropriate (see Supplementary Table 6). We used a permutation and resampling approach[100,101] detailed (detailed also in Supplementary Fig. 7). First, we calculated the Bhattacharyya distance between the two BTS histograms of interest as the metric for comparison. Next, we concatenated, permutated and then subsampled (10% for simulations, 20% for experiments) the transit time and brightness data. From this data, we generated BTS histograms and calculated the Bhattacharyya distance. This procedure was repeated 10,000 times (bootstrapping approach). The resulting Bhattacharyya distances (null statistics for null hypothesis that the original data originate from the same bi-variate distribution) were compared to the original Bhattacharyya distance for the two initial data sets. The $p$-value to reject the null hypothesis is then given by $\frac{\sum (BD_{null\_stats} > BD_{data}) + 1}{n_{it} + 1}$, where $BD_{null\_stats}$ is the vector containing the Bhattacharyya distances for the subsampled and permutated data, $BD_{data}$ is the Bhattacharyya distance for the original(input) histograms and $n_{it}$ is the number of iterations for the bootstrapping. The statistical analysis code in Python is available on GITHUB [https://github.com/Faldalf/sFCS_BTS].

All other statistical tests as indicated in the figure captions (e.g., t-tests or ANOVA) were performed using GraphPad Prism 10.

**Reporting summary**

Further information on research design is available in the Nature Portfolio Reporting Summary linked to this article.

## Data availability

Representative sFCS fitting results and example raw data are deposited on GITHUB. All raw sFCS data generated in this study have been deposited on Zenodo and are available under accession code 12747268. Source data for Fig. 1k–n, Fig. 2e, Fig. 3h, j are provided with this paper.

## Code availability

All custom code that has been used in this study is deposited on GITHUB and is associated with the following https://doi.org/10.5281/zenodo.12752907.

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

## Acknowledgements

We would like to thank the Kennedy Trust for Rheumatology Research (KTRR) and Engineering and Physical Sciences Research Council (EPSRC) awarded to the Rosalind Franklin Institute for funds in support of M.F. We thank the Wolfson Imaging Centre for providing the microscopy facility. We gratefully acknowledge the Oxford-ZEISS Centre of Excellence (Oxford-ZEISS CoE) in Biomedical Imaging for their support and assistance in this work. The Oxford-ZEISS CoE is supported by the Kennedy Trust for Rheumatology Research, IDRM and Carl Zeiss GMBH. M.F. and F.S. would like to thank the Wellcome Trust (212343/Z/18/Z) and EPSRC (EP/S004459/1). F.S. was supported by EMBO (ALTF 849-2020) and HFSP (LT000404/2021-L) fellowships supported by Scott E Frase. P.F.C. was supported by EMBO Long-Term Fellowship (ALTF 1420-2015, in conjunction with the European Commission (LTFCOFUND2013, GA-2013-609409 and Marie Sklodowska-Curie Actions) and Oxford-Bristol Myers Squibb Fellowship. M.L.D. also thanks the Wellcome Trust (100262/Z/12/Z), the European Commision (SYNECT AdG 670930) and the KTRR. P.F.C. and M.L.D. are also supported by the Chinese Academy of Medical Sciences Oxford Institute, CAMS Innovation Fund for Medical Sciences (CIFMS) funding 2018-I2M-2-002. Finally, we would like to thank Erdinc Sezgin for helpful discussions and kindly providing AF488-His$_6$.

## Author contributions

F.S. and M.F. conceived and designed the project with support by M.L.D.. F.S. performed the imaging and FCS acquisitions. P.F.C. designed and provided the CD40-AF488, anti-CD40 antibodies and CD40LG, and prepared the primary T cells, B cells and monocytes and calibrated and prepared cell-stimulating and control SLBs. N.K. acquired the BTS data on EGFP titration and anti-CD40-fab. F.S. with help of M.F. developed the analysis pipeline. F.S. and M.F. analysed the data and wrote the manuscript which was read and edited by all authors.

## Competing interests

The authors declare no competing interests.

## Additional information

**Supplementary information** The online version contains
supplementary material available at

Falk Schneider or Marco Fritzsche.

**Peer review information** *Nature Communications* thanks Valerica Raicu,
and the other, anonymous, reviewer(s) for their contribution to the peer
review of this work. A peer review file is available.

