## [Peer Review File · Nature Communications]

Reviewers' Comments:

Reviewer #1:

Remarks to the Author:

The authors thoroughly revised their manuscript to address the reviewers' comments provided in response to their original submission. They did so by adding and clarifying the text, providing additional figures and tables as well as expanding or revising the existing ones, and including additional references and discussion. I was also impressed by the amount of work they put into addressing the other reviewers' criticism. Based on these improvements, I feel that this is very well-executed and interesting research. I am happy to recommend publication, provided that the authors carefully consider the following comments.

The new supplementary Table T1 nicely summarizes the existing techniques for probing oligomerization and diffusion. There are some incorrect statements, and a few techniques are missing. Those should also be corrected or discussed in the introduction and discussion sections, as appropriate.

(i) It is stated that SpIDA can be used to determine fractions of oligomers. It does not appear that it can. Some versions of SpIDA usually give the average size of the oligomer but do not separate them into fractions.

(ii) FRET-FLIM cannot determine fractions of oligomers, since FLIM data analysis actually requires knowledge of oligomer size (or fractions thereof) in order to decide how many lifetimes to extract. It should be listed separately as "Can it be used to look at oligomerization."

(iii) On a more general note, the last column in the Table combines "Can determine fractions" with "mix of oligomers" (by the way, there is a typo). These are not equivalent, and techniques that can determine the average oligomer size for a mixture are usually distinct from those that can separate into fractions. These should be separated on the table.

(iv) A recent publication released after the initial submission of this manuscript (<https://www.biorxiv.org/content/10.1101/2024.03.18.585390v1>) shows that the combination of single molecule tracking and photobleaching steps provides diffusion coefficients separately for each oligomer size. This may not invalidate the novelty and analytical power of BTS, but it should not be overlooked.

Reviewer #2:

Remarks to the Author:

In the manuscript "Quantifying biomolecular organisation in membranes with brightness-transit statistics" the Authors Falk Schneider, Pablo F Cespedes, Michael L Dustin and Marco Fritzsche present a method called brightness-transit statistics (BTS) that relies on statistical analysis of data acquired using a standard confocal laser scanning microscope (CLSM) to simultaneously characterize molecular diffusion and oligomerization dynamics of biomolecules. While procedures for the analysis of the cellular dynamics of molecules acquired using CLSM systems are attractive for biomedical research since CLSM systems are nowadays widely available, important limitations were identified in the manuscript. Most notably, the writing is cryptic and open to interpretation, frequently leaving the reader to wonder about crucial details and sometimes even leading to misunderstandings. Additionally, key information is often missing, making it challenging to independently replicate the experiments described in the text. Due to this, I do not recommend the publication of this manuscript in its present form and suggest thorough revision of writing.

Please find my specific comments outlined below. Of note, this is not an exclusive list of comments. The Authors need to thoroughly revise their writing.

Specific concerns

Lines 156-158: It is stated: "The BTS data are acquired in the same way as a line scan (x_t) fluorescence fluctuation experiment wherein the same line of pixels is scanned 105 times." The large number of repetitions is needed to acquire good statistics for subsequent temporal autocorrelation analysis. What is missing is information on the pixel dwell time, i.e., scanning speed, and discussion on how to choose the best scanning speed, how is scanning speed related to photobleaching and what is the fastest diffusion that can be measured in relation to the scanning speed used.

Lines 163-165: It is stated: "We focus on the population level statistics from acquisitions over multiple conditions or cells and thus disregard the spatial information from each individual scan." What does this mean? Is one averaging the data acquired in different pixels along one line? Please revise this sentence or describe in detail how one is performing the analysis. Also, please, explain how the analysis "over multiple conditions" is performed.

Lines 165-167: It is stated: "Brightness can also be calculated using the first and second moment of the intensity distribution only (no correlation), this has, nevertheless, proven impractical due to the need for daily detector calibrations and sensitivity to photo-bleaching." This may be misunderstood to imply that temporal autocorrelation analysis is "insensitive" to photobleaching. Temporal autocorrelation analysis is indeed "sensitive" to photobleaching, and great care needs to be taken during the measurements to acquire the fluorescence intensity fluctuations with as little photobleaching as possible.

Lines 167-170: It is stated: "We employ this analysis on SLBs with and without interaction with immune cells to investigate changes in dynamic organisation, i.e., diffusion coefficient, oligomerisation and diffusion mode (Fig. 1a)". This statement is misleading. Fig. 1a is a cartoon.

Lines 195-200: It is difficult to understand the following sentence: "For this, we initially simulated a large ensemble of freely diffusing at different transit times ($\tau_{D,1}=1038.7$ ms, $\tau_{D,2}=103.9$ ms, $\tau_{D,3}= 20.8$ ms $\tau_{D,4}= 10.4$ ms, $\tau_{D,5}= 2.1$ ms, $\tau_{D,6}= 1.0$ ms) corresponding to diffusion coefficients ($D_1=0.01$ $\mu\text{m}^2/\text{s}$, $D_2=0.1$ $\mu\text{m}^2/\text{s}$, $D_3=0.5$ $\mu\text{m}^2/\text{s}$, $D_4=1$ $\mu\text{m}^2/\text{s}$, $D_5=5$ $\mu\text{m}^2/\text{s}$, and $D_6=10$ $\mu\text{m}^2/\text{s}$ assuming a Full Width at Half Maximum (FWHM) of 240 nm and a Gaussian observation volume) over three orders of magnitude, two different sFCS scanning sampling frequencies, and at three different molecular brightness (Fig. 1c,d and Supplementary Fig. S1)".

Please consider revising. For example: We have simulated a large ensemble of freely diffusing molecules moving at transit times (τ) that span over three orders of magnitudes: $\tau_{D,1}=1038.7$ ms, $\tau_{D,2}=103.9$ ms, $\tau_{D,3}= 20.8$ ms $\tau_{D,4}= 10.4$ ms, $\tau_{D,5}= 2.1$ ms, $\tau_{D,6}= 1.0$ ms; corresponding respectively to diffusion coefficients: $D_1=0.01$ $\mu\text{m}^2/\text{s}$, $D_2=0.1$ $\mu\text{m}^2/\text{s}$, $D_3=0.5$ $\mu\text{m}^2/\text{s}$, $D_4=1$ $\mu\text{m}^2/\text{s}$, $D_5=5$ $\mu\text{m}^2/\text{s}$, and $D_6=10$ $\mu\text{m}^2/\text{s}$, assuming a Full Width at Half Maximum (FWHM) of 240 nm and a Gaussian observation volume. Two different sFCS scanning sampling frequencies, 1000 Hz or 2000 Hz, were tested in the simulations (Supplementary Fig. S1) and three different molecular brightness values: 12.5 kHz, 25 kHz or 50 kHz. (Fig. 1c,d and Supplementary Fig. S1).

- In relation to the scanning sampling frequencies of 1000 Hz or 2000 Hz, how were these values chosen? And, how do they relate to the pixel dwell time that is usually given by instrument producers? For example, at our instrumental setup, the pixel dwell time could be varied from 0.64 $\mu\text{s}/\text{pixel}$ to 320 $\mu\text{s}/\text{pixel}$. What happens if a wider span of values for scanning sampling frequencies is chosen, e.g., for values between 500 and 5000 Hz?

- In relation to the Supplementary Fig. S1, please note that so-called normalized temporal autocorrelation curves are depicted. However, from the amplitude of the normalized temporal autocorrelation curve one cannot assess the brightness per molecule (cpm)! Also, it is not a good idea to self-correct the data. Rather, good measurements need to be performed. This needs to be corrected and better explained.

Lines 239-243: The statement: "We simulated multiple conditions, in which monomers organised into oligomers including dimers, trimers, and tetramers at various oligomerisation fractions ($f_1=0$, $f_2=0.05$, $f_3=0.1$, $f_4=0.2$, and $f_5=0.5$; meaning at f_5 half of all diffusing molecules are oligomers while the number of particles in the simulation is constant;" needs to be revised to clearly state that binary mixtures: monomers-dimers, monomers-trimers or monomer-tetramers, were analysed and that the contribution of the oligomeric fraction, f_i , increased as i increased from 1 to

5.

Lines 273-277: It is stated: "Visual inspection revealed no obvious qualitative differences in the resulting sFCS raw data or distribution of fluorescence intensity in confocal microscopy images (Supplementary Fig. S9 and S10), which was expected as even large oligomers are below the diffraction limit and the microscope's spatial resolution (Fig. 2b)." I respectfully disagree with this statement. One can clearly see in Supplementary Fig. S9 how the fluorescence intensity fluctuations change in the presented time series, and one nicely sees that the characteristic decay time of the temporal autocorrelation curves shifts towards longer lag times. A dashed vertical line that guides the eye, could be very helpful here. As for the later part of the sentence, I don't think that this is a useful comment and cannot see the link between the diffraction limit and the data shown in Fig. 2b. Either explain thoroughly or leave out this comment.

General comments

Please revise the text appearing on the figures. The text is often too small to be seen!
Please check your description in the figure legends and references in the text. More than once the content of the image and the description in the figure legend or in the text did not match.
Check the formulas.

Reviewer #3:

Remarks to the Author:

The Manuscript by Schneider et al. describes brightness-transit statistics (BTS) method that can assess diffusion dynamics and molecular oligomerization of biomolecules using scanning fluorescence correlation spectroscopy (sFCS) measurements. To validate the method, the Authors present detailed simulation results and evaluate the experimental systems in vitro and in the presence of live cells. Given that the method can be implemented with confocal microscopes and that typical measurement conditions and analysis/statistics codes are provided in GitHub, the BTS method may be useful for the general research community. Before the manuscript is ready for publication in Nature Communication, the Authors should either support their claims with the appropriate experimental system that generates true monomers or exact oligomeric species (e.g., dimers, tetramers) or more carefully word the conclusions from their experimental data (see below).

Comments:

The result section lists many diffusion processes BTS would be suitable for based on simulation ranges. Without experimental evidence, this section should be moved to discussion.

"Visual inspection revealed no obvious qualitative differences in the resulting sFCS raw data or distribution of fluorescence intensity in confocal microscopy images (Supplementary Fig. S9 and S10)," Perhaps use zoom-ins to show the point in Fig. S10. As is, these images do appear different.

"In addition, the results in the transitions from monomers to dimers and dimers to tetramers were confirmed with reductions in the average number of particles per micron squared (from 134 molecules/ μm^2 to 69 molecules/ μm^2 to 32 molecules/ μm^2 , respectively) when inferred from the sFCS acquisitions (Fig. 2d)."

Experimental reconstituted supported lipid bilayer system with eGFP clearly shows changes with the addition of primary and secondary antibodies. However, it cannot definitively produce only a specific type of oligomer (e.g., monomers, dimers, tetramers). Even the "monomeric control" is not demonstrated to be fully monomeric. While the authors mention assumptions in the discussion and the need to carefully characterize monomeric conditions, their experimental system is not ideal (e.g., EGFP is not a monomeric variant). A different system that indeed encompasses monomers or certain oligomers would be necessary for experimental validation. Alternatively, it should be noted in the text that results are suggestive of these oligomeric forms (as calculated by

the number of particles in sFCS).

Minor:

“Yet, a systematic methodology to quantify biomolecular organization remains elusive.”

Rephrase this sentence in the short abstract as it does not fully describe the state of the field as stated. In other parts of the manuscripts, this was properly communicated.

While the authors did significantly extend the introduction to describe the state of the field, reorganization would be beneficial for better flow.

Please provide a reference for FRET between GFP and AbberiorSTAR635P.

Reviewer #1

The authors thoroughly revised their manuscript to address the reviewers' comments provided in response to their original submission. They did so by adding and clarifying the text, providing additional figures and tables as well as expanding or revising the existing ones, and including additional references and discussion. I was also impressed by the amount of work they put into addressing the other reviewers' criticism. Based on these improvements, I feel that this is very well-executed and interesting research. I am happy to recommend publication, provided that the authors carefully consider the following comments.

We thank Reviewer #1 for their careful reading and recommendation to publish our work in Nature Communication. We are grateful for all the valuable comments that have improved our manuscript. We address the additional comments below.

The new supplementary Table T1 nicely summarizes the existing techniques for probing oligomerization and diffusion. There are some incorrect statements, and a few techniques are missing. Those should also be corrected or discussed in the introduction and discussion sections, as appropriate.

(i) It is stated that SpIDA can be used to determine fractions of oligomers. It does not appear that it can. Some versions of SpIDA usually give the average size of the oligomer but do not separate them into fractions.

We thank the Reviewer #1 for pointing this inaccuracy out which was corrected in the main text and in the Supplementary Table T1 in the revised manuscript. We fully agree that SpIDA can indeed not differentiate fraction of oligomers within one region of interest (ROI). However, similar to other techniques (also BTS or pCOMB) the SpIDA can estimate fractions for spatially segregated oligomers.

(ii) FRET-FLIM cannot determine fractions of oligomers, since FLIM data analysis actually requires knowledge of oligomer size (or fractions thereof) in order to decide how many lifetimes to extract. It should be listed separately as "Can it be used to look at oligomerization."

We updated the Supplementary Table T1 in the revised manuscript.

(iii) On a more general note, the last column in the Table combines "Can determine fractions" with "mix of oligomers" (by the way, there is a typo). These are not equivalent, and techniques that can determine the average oligomer size for a mixture are usually distinct from those that can separate into fractions. These should be separated on the table.

We thank the Reviewer #1 for their suggestion and revised the Supplementary Table T1 accordingly in the revised manuscript. To this end, the table now displays three columns to describe the different facets of oligomerization measurements:

"Can it determine fractions of different oligomers?",
"Can it determine fractions of different oligomers?", and
"How does it deal with a mixture of oligomers and monomers?"

(iv) A recent publication released after the initial submission of this manuscript (<https://www.biorxiv.org/content/10.1101/2024.03.18.585390v1>) shows that the combination of single molecule tracking and photobleaching steps provides diffusion coefficients separately for each oligomer size. This may not invalidate the novelty and analytical power of BTS, but it should not be overlooked.

We thank the Reviewer #1 for bringing this new preprint to our attention which we cited in the revised manuscript. The authors nicely show how a novel combination of techniques (SPT and single molecule photo-bleaching step analysis) can be used to illuminate molecular organization.

Importantly, the manuscript very nicely highlights the need for the BTS approach as their analysis was performed on different cells (fixed and live) and at very low expression levels.

Reviewer #2 (Remarks to the Author):

In the manuscript "Quantifying biomolecular organisation in membranes with brightness-transit statistics" the Authors Falk Schneider, Pablo F Cespedes, Michael L Dustin and Marco Fritzsche present a method called brightness-transit statistics (BTS) that relies on statistical analysis of data acquired using a standard confocal laser scanning microscope (CLSM) to simultaneously characterize molecular diffusion and oligomerization dynamics of biomolecules. While procedures for the analysis of the cellular dynamics of molecules acquired using CLSM systems are attractive for biomedical research since CLSM systems are nowadays widely available, important limitations were identified in the manuscript. Most notably, the writing is cryptic and open to interpretation, frequently leaving the reader to wonder about crucial details and sometimes even leading to misunderstandings. Additionally, key information is often missing, making it challenging to independently replicate the experiments described in the text. Due to this, I do not recommend the publication of this manuscript in its present form and suggest thorough revision of writing.

Please find my specific comments outlined below. Of note, this is not an exclusive list of comments. The Authors need to thoroughly revise their writing.

We thank the Reviewer #2 for their reading and overall positive assessment in favour of publication in Nature Communication in principle. Especially, the acknowledgment of its broad value for the biomedical community is very encouraging as highlighted in the previous revision cycle. We are also grateful that any technical points raised in revision in response to the review in Nature Methods are no longer of concern as they have not been commented on by the referee.

Nevertheless, the Reviewer #2 now raises multiple new concerns about the textual presentation and interpretation of the BTS method which have not been raised in the previous revision cycles. The referee points out six sentences that could possibly be misunderstood and therefore suggests to rephrase these sentences prior their recommendation for publication which has now been successfully addressed in the revised manuscript.

The referee requests further technical descriptions which in their opinion are missing to reproduce BTS experiments. We were surprised about this comment as we provide all details on experimental acquisition parameters in the Materials and Methods, Supplementary Information, and deposited files on Github that allow to reuse the acquisition settings on similar microscopes. However, this may be a misunderstanding and the referee may have overseen the detailed descriptions provided in the original manuscript as well as in the previous revision cycles. Importantly, because this concern was not raised in the previous revision and is also not shared by the Reviewer #1 and #3, the authors were unsure how to satisfactorily address this point. To still provide an effort to address this point, all authors and selected senior colleagues actively working in the field have again carefully helped to simplify the presentation in the revised manuscript. Finally, we have provided additional material in response to this concern that clarify the choice of the acquisition parameters in the manuscript representing gold standards in the FCS field. Please see our detailed comments below:

Specific concerns

Lines 156-158: It is stated: “The BTS data are acquired in the same way as a line scan (xt) fluorescence fluctuation experiment wherein the same line of pixels is scanned 10^5 times.” The large number of repetitions is needed to acquire good statistics for subsequent temporal autocorrelation analysis. What is missing is information on the pixel dwell time, i.e., scanning speed, and discussion on how to choose the best scanning speed, how is scanning speed related to photobleaching and what is the fastest diffusion that can be measured in relation to the scanning speed used.

We would like to respectfully refer the referee to the Materials and Method section in the original manuscript which presents an exact technical description of the measurements including the above requested pixel dwell time and scanning frequency.

A detailed description of the computer-simulations presented in Figure 1c-1f and Supplementary Figure S2-S4 is indeed provided in the main text for a range of diffusion coefficients.

For a more detailed explanation of an optimal choice of scanning frequency, as discussed previously, we refer to prominent literature which determines the temporal resolution of the measurement and line-time (inverse scanning frequency) should at least 5-10 times smaller than the measured transit time (PMID: 30028588, PMID: 16782786, PMID: 23521662, PMID: 34494547). We also reference now a detailed protocol with general rules and guidelines to optimize sFCS acquisitions (doi: 10.1007/978-1-0716-3135-5_5) in the Materials and Methods of the revised manuscript.

Finally, we would like to refer the referee to the previous revision cycle in which the photobleaching has been raised and addressed. We reiterate the discussion on reduced photobleaching due to the scanning nature in sFCS versus pFCS in the revised version of the manuscript. Furthermore, we discuss that photobleaching is at best avoided and otherwise needs to be cautiously corrected.

Lines 163-165: It is stated: “We focus on the population level statistics from acquisitions over multiple conditions or cells and thus disregard the spatial information from each individual scan.” What does this mean? Is one averaging the data acquired in different pixels along one line? Please revise this sentence or describe in detail how one is performing the analysis. Also, please, explain how the analysis “over multiple conditions” is performed.

We simplified the sentence following the suggestion of the referee in the revised manuscript. To further explain the approach, fitting parameters over multiple measurements are pooled to obtain the BTS histogram for one condition. This can be compared to the histogram from another condition (eg., control versus treated). Note, we do not make use of the spatial information in the scan, for example, we do not differentiate centre vs periphery of a cell (Please also check Schneider et al, ACS Nano 2018).

Lines 165-167: It is stated: “Brightness can also be calculated using the first and second moment of the intensity distribution only (no correlation), this has, nevertheless, proven impractical due to the need for daily detector calibrations and sensitivity to photo-bleaching.” This may be misunderstood to imply that temporal autocorrelation analysis is “insensitive” to photobleaching. Temporal autocorrelation analysis is indeed “sensitive” to photobleaching, and great care needs to be taken during the measurements to acquire the fluorescence intensity fluctuations with as little photobleaching as possible.

We thank the referee for pointing the possible misunderstanding out. The sentence has been rephrased in the revised manuscript. Because we strongly agree with the referee that photo-bleaching is an important issue to be addressed, we have further extended the discussion around photo-bleaching in the revised manuscript.

Lines 167-170: It is stated: “We employ this analysis on SLBs with and without interaction with immune cells to investigate changes in dynamic organisation, i.e., diffusion coefficient, oligomerisation and diffusion mode (Fig. 1a)”. This statement is misleading. Fig. 1a is a cartoon.

We clarified that the experimental setup is sketched in form of a cartoon in Figure 1a.

Lines 195-200: It is difficult to understand the following sentence: “For this, we initially simulated a large ensemble of freely diffusing at different transit times ($\tau_{D,1}=1038.7$ ms, $\tau_{D,2}=103.9$ ms, $\tau_{D,3}=20.8$ ms $\tau_{D,4}=10.4$ ms, $\tau_{D,5}=2.1$ ms, $\tau_{D,6}=1.0$ ms) corresponding to diffusion coefficients ($D_1=0.01$ $\mu\text{m}^2/\text{s}$, $D_2=0.1$ $\mu\text{m}^2/\text{s}$, $D_3=0.5$ $\mu\text{m}^2/\text{s}$, $D_4=1$ $\mu\text{m}^2/\text{s}$, $D_5=5$ $\mu\text{m}^2/\text{s}$, and $D_6=10$ $\mu\text{m}^2/\text{s}$ assuming a Full Width at Half Maximum (FWHM) of 240 nm and a Gaussian observation volume) over three orders of magnitude, two different sFCS scanning sampling frequencies, and at three different molecular brightness (Fig. 1c,d and Supplementary Fig. S1)”. Please consider revising. For example: We have simulated a large ensemble of freely diffusing molecules moving at transit times (τ) that span over three orders of magnitudes: $\tau_{D,1}=1038.7$ ms, $\tau_{D,2}=103.9$ ms, $\tau_{D,3}=20.8$ ms $\tau_{D,4}=10.4$ ms, $\tau_{D,5}=2.1$ ms, $\tau_{D,6}=1.0$ ms; corresponding respectively to diffusion coefficients: $D_1=0.01$ $\mu\text{m}^2/\text{s}$, $D_2=0.1$ $\mu\text{m}^2/\text{s}$, $D_3=0.5$ $\mu\text{m}^2/\text{s}$, $D_4=1$ $\mu\text{m}^2/\text{s}$, $D_5=5$ $\mu\text{m}^2/\text{s}$, and $D_6=10$ $\mu\text{m}^2/\text{s}$, assuming a Full Width at Half Maximum (FWHM) of 240 nm and a Gaussian observation volume. Two different sFCS scanning sampling frequencies, 1000 Hz or 2000 Hz, were tested in the simulations (Supplementary Fig. S1) and three different molecular brightness values: 12.5 kHz, 25 kHz or 50 kHz. (Fig. 1c,d and Supplementary Fig. S1).

We rephrased the sentence in the revised manuscript.

- In relation to the scanning sampling frequencies of 1000 Hz or 2000 Hz, how were these values chosen? And, how do they relate to the pixel dwell time that is usually given by instrument producers? For example, at our instrumental setup, the pixel dwell time could be varied from 0.64 $\mu\text{s}/\text{pixel}$ to 320 $\mu\text{s}/\text{pixel}$. What happens if a wider span of values for scanning sampling frequencies is chosen, e.g., for values between 500 and 5000 Hz?

We chose the values according to our instrument (ZEISS LSM 780) that had a maximum scanning frequency of 2069 Hz. Slower acquisition speeds (1000 Hz and lower) reduce the SNR over the tested parameter space (as shown in Supplementary Figure S3). As discussed above (first comment of the reviewer), the scanning frequency determines the temporal resolution (PMID: 30028588, PMID: 16782786, PMID: 23521662, PMID: 34494547). The pixel dwell time at our instrument is a function of the zoom (pixel size) and scanning frequency. Newer instruments (such as the ZEISS LSM 980) allow for higher sampling rates (up to ~ 3400 Hz).

The scanning frequency should be chosen according to the dynamic range of diffusion coefficients expected in the measurement (see Materials and Methods section). While faster scanning might help with SNR, at ZEISS instruments the maximum number of

lines is fixed limiting the total acquisition time. Therefore, when studying slow dynamics ($<0.1 \text{ um}^2/\text{s}$), it might be advisable to reduce scan speed and thus increase acquisition time.

- In relation to the Supplementary Fig. S1, please note that so-called normalized temporal autocorrelation curves are depicted. However, from the amplitude of the normalized temporal autocorrelation curve one cannot assess the brightness per molecule (cpm)! Also, it is not a good idea to self-correct the data. Rather, good measurements need to be performed. This needs to be corrected and better explained.

We totally agree with the referee that the cpm cannot be obtained from a normalized ACF. The amplitude of the depicted simulation was coincidentally around 1 (the y axis clearly states “Correlation”, not “Normalized Correlation”). To avoid possible confusion, we changed the scale on the y-axis of the correlation curve plot and indicated on the correlation carpet “Norm. Correlation”. Furthermore, we mention on the workflow that it is good practice to keep post-processing and corrections to a minimum and optimize data acquisition.

Lines 239-243: The statement: “We simulated multiple conditions, in which monomers organised into oligomers including dimers, trimers, and tetramers at various oligomerisation fractions ($f_1=0$, $f_2=0.05$, $f_3=0.1$, $f_4=0.2$, and $f_5=0.5$; meaning at f_5 half of all diffusing molecules are oligomers while the number of particles in the simulation is constant;” needs to be revised to clearly state that binary mixtures: monomers-dimers, monomers-trimers or monomer-tetramers, were analysed and that the contribution of the oligomeric fraction, f_i , increased as i increased from 1 to 5.

We rephrased the sentence in the revised manuscript.

Lines 273-277: It is stated: “Visual inspection revealed no obvious qualitative differences in the resulting sFCS raw data or distribution of fluorescence intensity in confocal microscopy images (Supplementary Fig. S9 and S10), which was expected as even large oligomers are below the diffraction limit and the microscope’s spatial resolution (Fig. 2b).” I respectfully disagree with this statement. One can clearly see in Supplementary Fig. S9 how the fluorescence intensity fluctuations change in the presented time series, and one nicely sees that the characteristic decay time of the temporal autocorrelation curves shifts towards longer lag times. A dashed vertical line that guides the eye, could be very helpful here. As for the later part of the sentence, I don’t think that this is a useful comment and cannot see the link between the diffraction limit and the data shown in Fig. 2b. Either explain thoroughly or leave out this comment.

We clarified the sentence and added a dashed line to the figure in the revised manuscript.

General comments

Please revise the text appearing on the figures. The text is often too small to be seen!

Standard fonts and font sizes were chosen for all PDF figures typical for publication in Nature Communications. We hope for guidance by the Editorial team if further changes are required.

Please check your description in the figure legends and references in the text. More than once the content of the image and the description in the figure legend or in the text did not match.

All authors have carefully read the figure legends and references again and could not find any mistakes.

Check the formulas.

All authors have carefully read all equations again and rectified any mistakes.

Reviewer #3

The Manuscript by Schneider et al. describes brightness-transit statistics (BTS) method that can assess diffusion dynamics and molecular oligomerization of biomolecules using scanning fluorescence correlation spectroscopy (sFCS) measurements. To validate the method, the Authors present detailed simulation results and evaluate the experimental systems in vitro and in the presence of live cells. Given that the method can be implemented with confocal microscopes and that typical measurement conditions and analysis/statistics codes are provided in GitHub, the BTS method may be useful for the general research community. Before the manuscript is ready for publication in Nature Communication, the Authors should either support their claims with the appropriate experimental system that generates true monomers or exact oligomeric species (e.g., dimers, tetramers) or more carefully word the conclusions from their experimental data (see below).

We thank Reviewer #3 for their overall positive assessment and recommendation to publish our work in Nature Communication in principle. The specific points are addressed below.

Comments:

The result section lists many diffusion processes BTS would be suitable for based on simulation ranges. Without experimental evidence, this section should be moved to discussion.

We moved the biological examples for different diffusion time-scales to the discussion section in the revised manuscript.

“Visual inspection revealed no obvious qualitative differences in the resulting sFCS raw data or distribution of fluorescence intensity in confocal microscopy images (Supplementary Fig. S9 and S10),”

Perhaps use zoom-ins to show the point in Fig. S10. As is, these images do appear different.

We amended the statement indicating slight changes to the distribution of fluorescence intensity and added the zoom-ins to Supplementary Figure S10 in the revised manuscript.

“In addition, the results in the transitions from monomers to dimers and dimers to tetramers were confirmed with reductions in the average number of particles per micron squared (from 134 molecules/ μm^2 to 69 molecules/ μm^2 to 32 molecules/ μm^2 , respectively) when inferred from the sFCS acquisitions (Fig. 2d).” Experimental reconstituted supported lipid bilayer system with eGFP clearly shows changes with the addition of primary and secondary antibodies. However, it cannot definitively produce only a specific type of oligomer (e.g., monomers, dimers, tetramers). Even the “monomeric control” is not demonstrated to be fully monomeric. While the authors mention assumptions in the discussion and the need to carefully characterize monomeric conditions, their experimental system is not ideal (e.g., EGFP is not a monomeric variant). A different system that indeed encompasses monomers or certain oligomers would be necessary for experimental validation. Alternatively, it should be noted in the text that results are suggestive of these oligomeric forms (as calculated by the number of particles in sFCS).

We rectified the wording in the Results section of the revised version of the manuscript. Furthermore, we added additional discussion to the Discussion section in the revised manuscript.

Minor:

“Yet, a systematic methodology to quantify biomolecular organization remains elusive.”

Rephrase this sentence in the short abstract as it does not fully describe the state of the field as stated. In other parts of the manuscripts, this was properly communicated.

We rephrased that sentence in the same way as in the abstract in the revised manuscript:

“Yet, a methodology to systematically quantify biomolecular organisation, measuring diffusion dynamics and oligomerisation, represents an unmet need.”

While the authors did significantly extend the introduction to describe the state of the field, reorganization would be beneficial for better flow.

We did a final revision of the introduction of the revised manuscript.

Please provide a reference for FRET between GFP and AbberiorSTAR635P.

We added the reference (PMID: 33364580) to the main text of the revised manuscript.

Reviewers' Comments:

Reviewer #1:

Remarks to the Author:

The authors have fully addressed all of my comments and suggestions. This is a nice piece of work. Congratulations!

Reviewer #2:

Remarks to the Author:

The authors have addressed most of my concerns. I have several minor suggestions, as indicated below, and do not need to see the revised manuscript again.

Page 5, line 171: Please remove "for experimental setup".

Page 5, line 173, reference to Supplementary Table T1: The statement in the last two columns in the second row of Supplementary Table T1 is not correct. While there are limitations, FCS can be used to determine the fractions of different oligomers in a mixture. See for example: Thompson, N. L. (1991). Fluorescence Correlation Spectroscopy. Topics in Fluorescence Spectroscopy. J. R. Lakowicz. New York, Plenum Press. 1: 337-378.

Page 8, line 263: Please remove "depicting experimental setup".

Page 13, line 448: Delete "is thought to". It is well established that fast scanning induces less photobleaching than slow scanning/immobile laser beam.

Page 14, line 476: Correct typo: "ie." To "i.e.,".

Page 19, line 671: Abbreviation "pFCS" is not defined. It can be added on page 13, line 449 where "point FCS" was first mentioned.

Page 21, lines 753-754: Please revise the sentence: "The first 10 s of each measurement were cropped off to remove immobile molecules." Obviously, "immobile molecules" cannot be removed by "cropping off the first 10 s of each measurement". What the Authors want to say is: "To minimize potential temporal autocorrelation curve distortion due to photobleaching of immobile/slowly moving components, the first 10 s of each measurement were left out and temporal autocorrelation analysis was performed on the truncated time series."

Page 25, Figure 1 g and h: Please disregard this comment if it is due to my false impression. It seems to me that the change in transit time is larger than the expected $\sqrt[3]{2} = 1.26$ for monomer-dimer and $\sqrt[3]{3} = 1.44$ for monomer-trimer. I assume that monomer transit time is 10.39 ms ($\ln 10.39 = 2.34$), in which case I expect the dimer transit time to be 2.57 and the trimer transit time 2.71. Transit times for 10 % contribution of dimer/trimer appear to be longer than for pure dimer/trimer.

Page 26, line 847: Full stop is missing after "(right)".

Page 26, line 862: Please replace "and" with "or" to facilitate understanding that binary mixtures of monomers with dimers (l), trimers (m) or tetramers (n) were simulated.

My answers to the specific questions above are given below.

1. What are the noteworthy results?

A method for analyzing fluorescence intensity fluctuations acquired by line scanning fluorescence correlation spectroscopy (FCS) is presented. The theoretical concepts are verified by numerical simulations and the usefulness of this tool for biomedical research is demonstrated in two in vitro applications aiming to characterize biomolecular organization in model membranes, one not involving and one involving live cells. The simulation and analysis code could be useful for the wider biomed community.

2. Will the work be of significance to the field and related fields? How does it compare to the established literature? If the work is not original, please provide relevant references. The main value of this work lies in the possibility to widely adopt this analytical method in biomedical research. In essence, all laboratories equipped with a confocal laser scanning microscope of a recent generation could adopt it in their studies of molecular mechanisms underlying normal and disease physiology. The concept is not new - conventional, stationary-beam or point FCS and scanning FCS have both been used before to characterize molecular organization in model membranes and in live cell plasma membranes. See for example:

Petrásek Z, Ries J, Schuille P. Scanning FCS for the characterization of protein dynamics in live cells. *Methods Enzymol.* 2010;472:317-43. doi: 10.1016/S0076-6879(10)72005-X. PMID: 20580970.

Clark NM, Sozzani R. Measuring Protein Movement, Oligomerization State, and Protein-Protein Interaction in Arabidopsis Roots Using Scanning Fluorescence Correlation Spectroscopy (Scanning FCS). *Methods Mol Biol.* 2017;1610:251-266. doi: 10.1007/978-1-4939-7003-2_16. PMID: 28439868.

Nevertheless, the simulation and analysis code could be useful.

3. Does the work support the conclusions and claims, or is additional evidence needed? The work supports to a very large extent the conclusions and claims. Additional evidence is not needed. However, methodology limitations could be expounded on.

4. Are there any flaws in the data analysis, interpretation and conclusions? Do these prohibit publication or require revision?

There are no flows identified, but the method has the same challenges as other FCS-based methods. Nevertheless, the simulation and analysis code could be useful.

5. Is the methodology sound? Does the work meet the expected standards in your field? The methodology is sound but limitations also exist. The simulation and analysis code could be useful.

6. Is there enough detail provided in the methods for the work to be reproduced? Yes, to a very large extent. I have, however, not checked the code.

Reviewer #3:

Remarks to the Author:

The Authors have addressed all my concerns. The manuscript is significantly improved.

Final Point-by-Point Responses to Reviewer's comments

Reviewer #1 (Remarks to the Author):

The authors have fully addressed all of my comments and suggestions. This is a nice piece of work. Congratulations!

We thank Reviewer #1 for their comments and help with improving our manuscript.

Reviewer #2 (Remarks to the Author):

The authors have addressed most of my concerns. I have several minor suggestions, as indicated below, and do not need to see the revised manuscript again.

We thank Reviewer #2 for their comments and address the specific points below as well as in the revised manuscript.

Page 5, line 171: Please remove “for experimental setup”.

The point has been addressed in the revised manuscript.

Page 5, line 173, reference to Supplementary Table T1: The statement in the last two columns in the second row of Supplementary Table T1 is not correct. While there are limitations, FCS can be used to determine the fractions of different oligomers in a mixture. See for example:

Thompson, N. L. (1991). Fluorescence Correlation Spectroscopy. Topics in Fluorescence Spectroscopy. J. R. Lakowicz. New York, Plenum Press. 1: 337-378.

We agree with the referee on the theoretical potential of the suggested methodology to resolve oligomers and that there is a broad recognition of the difficulties in the field which limit the applications. We indicate this clearly in Supplementary Table 1 of the revised manuscript.

Although these and other authors demonstrate the potential to detect oligomers in a mixture using pure monomer/oligomer solutions or higher order correlation functions (higher order cumulants), their own experiments show practical limitations in solution experiments. An extension to measurements in living cells in the presence of significantly higher noise and considering the slower diffusion dynamics in membranes would require very long acquisition times potentially resulting in high photobleaching and suffer from issues with cellular movements, making these measurements practically very challenging to study organization at the plasma membrane.

Page 8, line 263: Please remove “depicting experimental setup”.

The point has been addressed in the revised manuscript.

Page 13, line 448: Delete “is thought to”. It is well established that fast scanning induces less photobleaching than slow scanning/immobile laser beam.

The point has been addressed in the revised manuscript.

Page 14, line 476: Correct typo: “ie.” To “i.e.”.

The point has been addressed in the revised manuscript.

Page 19, line 671: Abbreviation “pFCS” is not defined. It can be added on page 13, line 449 where “point FCS” was first mentioned.

The point has been addressed in the revised manuscript.

Page 21, lines 753-754: Please revise the sentence: “The first 10 s of each measurement were cropped off to remove immobile molecules.” Obviously, “immobile molecules” cannot be removed by “cropping off the first 10 s of each measurement”. What the Authors want to say is: “To minimize potential temporal autocorrelation curve distortion due to photobleaching of immobile/slowly moving components, the first 10 s of each measurement were left out and temporal autocorrelation analysis was performed on the truncated time series.”

The point has been addressed in the revised manuscript.

Page 25, Figure 1 g and h: Please disregard this comment if it is due to my false impression. It seems to me that the change in transit time is larger than the expected $\sqrt[3]{2} = 1.26$ for monomer-dimer and $\sqrt[3]{3} = 1.44$ for monomer-trimer. I assume that monomer transit time is 10.39 ms ($\ln 10.39 = 2.34$), in which case I expect the dimer transit time to be 2.57 and the trimer transit time 2.71. Transit times for 10 % contribution of dimer/trimer appear to be longer than for pure dimer/trimer.

The reviewer may be under the false impression that we are investigating 3D diffusion. For this case, the reviewer would be correct. However, our article is focused on 2D diffusion, and we simulated membrane-bound molecules. We outline the underlying assumptions (as related to Saffman–Delbrück model) for causing the slow-downs during oligomerization in the Methods section in detail.

Page 26, line 847: Full stop is missing after “(right)”.

The point has been addressed in the revised manuscript.

Page 26, line 862: Please replace “and” with “or” to facilitate understanding that binary mixtures of monomers with dimers (l), trimers (m) or tetramers (n) were simulated.

The point has been addressed in the revised manuscript.

My answers to the specific questions above are given below.

1. What are the noteworthy results?

A method for analyzing fluorescence intensity fluctuations acquired by line scanning fluorescence correlation spectroscopy (FCS) is presented. The theoretical concepts are verified by numerical simulations and the usefulness of this tool for biomedical research is demonstrated in two in vitro applications aiming to characterize biomolecular organization in model membranes, one not involving and one involving live cells. The simulation and analysis code could be useful for the wider biomed community.

2. Will the work be of significance to the field and related fields? How does it compare to the established literature? If the work is not original, please provide relevant references.

The main value of this work lies in the possibility to widely adopt this analytical method in biomedical research. In essence, all laboratories equipped with a confocal laser scanning microscope of a recent generation could adopt it in their studies of molecular mechanisms underlying normal and disease physiology. The concept is not new - conventional, stationary-beam or point FCS and scanning FCS have both been used before to characterize molecular organization in model membranes and in live cell plasma membranes. See for example:

Petrásek Z, Ries J, Schwille P. Scanning FCS for the characterization of protein dynamics in live cells. *Methods Enzymol.* 2010;472:317-43. doi: 10.1016/S0076-6879(10)72005-X. PMID: 20580970.

Clark NM, Sozzani R. Measuring Protein Movement, Oligomerization State, and Protein-Protein Interaction in Arabidopsis Roots Using Scanning Fluorescence Correlation Spectroscopy (Scanning FCS). *Methods Mol Biol.* 2017;1610:251-266. doi: 10.1007/978-1-4939-7003-2_16. PMID: 28439868.

Nevertheless, the simulation and analysis code could be useful.

3. Does the work support the conclusions and claims, or is additional evidence needed?

The work supports to a very large extent the conclusions and claims. Additional evidence is not needed. However, methodology limitations could be expounded on.

4. Are there any flaws in the data analysis, interpretation and conclusions? Do these prohibit publication or require revision?

There are no flaws identified, but the method has the same challenges as other FCS-based methods. Nevertheless, the simulation and analysis code could be useful.

5. Is the methodology sound? Does the work meet the expected standards in your field?

The methodology is sound but limitations also exist. The simulation and analysis code could be useful.

6. Is there enough detail provided in the methods for the work to be reproduced?

Yes, to a very large extent. I have, however, not checked the code.

Reviewer #3 (Remarks to the Author):

The Authors have addressed all my concerns. The manuscript is significantly improved.

We thank Reviewer #2 for their comments and help with improving our manuscript.